# Dissecting the basis for differential substrate specificity of ADAR1 and ADAR2

Marlon S. Zambrano-Mila[1], Monika Witzenberger [1], Zohar Rosenwasser[2], Anna Uzonyi[1], Ronit Nir[1], Shay Ben-Aroya[2], Erez Y. Levanon [2] & Schraga Schwartz[1] ✉

Millions of adenosines are deaminated throughout the transcriptome by ADAR1 and/or ADAR2 at varying levels, raising the question of what are the determinants guiding substrate specificity and how these differ between the two enzymes. We monitor how secondary structure modulates ADAR2 vs ADAR1 substrate selectivity, on the basis of systematic probing of thousands of synthetic sequences transfected into cell lines expressing exclusively ADAR1 or ADAR2. Both enzymes induce symmetric, strand-specific editing, yet with distinct offsets with respect to structural disruptions: −26 nt for ADAR2 and −35 nt for ADAR1. We unravel the basis for these differences in offsets through mutants, domain-swaps, and ADAR homologs, and find it to be encoded by the differential RNA binding domain (RBD) architecture. Finally, we demonstrate that this offset-enhanced editing can allow an improved design of ADAR2-recruiting therapeutics, with proof-of-concept experiments demonstrating increased on-target and potentially decreased off-target editing.

Millions of adenosines are deaminated into inosines transcriptome-wide[1,2], catalyzed by two deaminating enzymes, ADAR1 (ADAR) and ADAR2 (ADARB1). Inosine is perceived as guanosine by the internal cellular machinery, and hence editing can result in protein recoding[3–7], alternative splicing[8–10], and alterations in targeting and maturation of microRNA[11,12]. In parallel, editing can also alter the RNA secondary structure, and in doing so modulate the immunogenicity of self and viral RNAs within cells[13–17]. In accordance with the wide distribution of edited sites, abnormal dysregulation of A-to-I deamination has been associated with a broad spectrum of human diseases[18], and targeting of ADAR enzymes is an emerging therapeutic strategy in cancer[19].

Different adenosines throughout the transcriptome are edited at dramatically different efficiencies (or not at all), begging the question of what governs enzymatic selectivity towards specific targets. Understanding the rules guiding these two enzymes to their diverse targets is of intense interest not only from biological and pathological perspectives, but also from therapeutic ones. In recent years, unraveling the rules dictating deamination via the ADAR enzymes has accrued substantial interest in the context of ongoing efforts to achieve targeted mRNA editing. Targeted editing is emerging as a therapeutic modality that may potentially offer a safer alternative to correct single-nucleotide mutations[20] in comparison to CRISPR-mediated DNA editing. Diverse approaches have been implemented in recent years to recruit ADAR enzymes towards specific substrates[21–25]. Although successful, these attempts often resulted only in partial efficiencies and in some cases also with considerable off-target effects[26]. Improving our understanding of the rules guiding inosine formation and of the factors determining enzyme specificity will pave the path toward the development of both more optimal editors and improved guides.

Studies exploring the targeting efficiencies of ADAR1 and ADAR2 have revealed several general principles. First, the specificity of these two enzymes is only partially overlapping[27,28], suggesting differences in the selectivity of these two enzymes. Second, RNA secondary structure plays a critical role. ADAR1 targets are nearly inevitably within long double-stranded RNAs[29], and hence highly enriched in repetitive elements such as Alu and long interspersed elements[30–32]. ADAR2 targets tend to be in duplex regions interrupted by mismatches or loops[33–36], and in-vitro work has shown that distal bulges can, at times, impact editing efficiency[34]. Yet, the structural rules governing editing—which

[1]Department of Molecular Genetics, Weizmann Institute of Science, Rehovot 7630031, Israel. [2]Faculty of Life Sciences, Bar Ilan University, 5290002 Ramat Gan, Israel. ✉e-mail: schwartz@weizmann.ac.il

would be of critical importance for predictive models—are not understood. In addition, for both ADAR1 and ADAR2, A-C mismatch harboring targets are particularly prone to undergo editing[37]. Finally, the sequence also plays a role in target selectivity. In-vitro editing assays with artificial RNA duplexes revealed that ADAR1 and ADAR2 prefer to edit adenosines depleted of G's at the position preceding the target and show some bias for a G downstream of the target edited site[35,38,39].

The factors underlying the differences in specificity between ADAR1 and ADAR2 are understood only to a limited extent. These two paralog proteins, which likely evolved via a gene duplication event roughly 700 million years ago[40], differ in their protein domain architecture. The catalytic domain, present on both ADARs, was shown to play a role in the definition of selectivity[37,41]. In addition to the catalytic domain, human ADAR1 contains either one or two Zα domains (dependent on the isoform) and three RNA binding domains (RBDs) whereas human ADAR2 contains two RBDs but no Zα domains. The RBDs participate in dsRNA substrate recognition and RNA binding[42], and were suggested to partially mediate ADAR selectivity via both sequence-specific and non-specific mechanisms[34,43]. The Z domains, binding left-handed nucleic acids, have been implicated in allowing co-transcriptional binding of ADAR1 to nascent RNAs[44,45] and more recently in preventing Z-RNA dependent activation of pathogenic interferon by Z-DNA binding protein 1 (ZBP1)[46–48]. Whether they have a role in defining substrate specificity is unclear.

To systematically dissect how substrate selectivity by ADAR1 is governed by secondary structure, we previously screened ADAR1-mediated editing across thousands of sequence variants, which had been designed to systematically perturb the secondary structure along two highly double-stranded backbones. We discovered that the introduction of structural disruptions within an otherwise perfect double-stranded RNA structure gives rise to robust and predictable ADAR1-mediated editing at a fixed offset of 35 base pairs (bp) upstream from the disruption[49]. Whether structural disruptions of ADAR2 targets lead to editing at a fixed offset, and what the mechanistic basis for this offset is, remains unknown.

Here, we systematically monitor how secondary structure modulates ADAR2 substrate selectivity, on the basis of systematic probing of thousands of synthetic sequences transfected into ADAR1-deleted cell lines exogenously expressing ADAR2. We find that similarly to ADAR1, structural disruptions give rise to symmetric, strand-specific induced editing at a fixed offset. However, in contrast to ADAR1 acting at a −35 bp offset, in the case of ADAR2, structural disruptions give rise to induced editing at an offset of −26 bp. We dissect the basis for the differences in offset between ADAR1 and ADAR2 via diverse mutants, domain-swaps and ADAR evolutionary homologs. We uncover that the difference in the offset is encoded by the differential RNA binding domain architecture of the two ADARs, yet that it is not determined by the number of RBDs. We demonstrate that this understanding of ADAR2 specificity can allow an improved design of ADAR2-recruiting therapeutics, yielding increased on-target editing, with some evidence also for reduced off-target editing. Our findings provide comprehensive insight into the features determining ADAR2 substrate selectivity and into the roles of the RNA binding domains of ADAR1 and ADAR2 in mediating differential targeting, and should facilitate the design of improved ADAR2-recruiting therapeutics.

## Results

### Screening of ADAR2 substrates

We sought to systematically compare the targeting specificity of ADAR2 to its ADAR1 counterpart. Toward this goal, we employed a pool of thousands of sequence variants that we had previously designed to probe the specificity of ADAR1, described in[49]. In brief, these sequence variants are based on two distinct backbones folding into a perfect hairpin structure: the endogenous mouse B2 element,

serving as a more natural editing target, and a sequence complementary to the 3' UTR of the fluorescent reporter mNeonGreen (mNG) transcript, serving as a completely synthetic target. In both cases, this hairpin consists of a 146-nt long stem and a 46-nt long loop (Fig. 1A). For each of these two backbones, we previously designed and synthesized roughly two thousand sequence variants systematically perturbing the hairpin structure via random structural disruptions, systematic incorporation of single, double, or random mismatches, the introduction of pyrimidine-rich bulges, and systematic shortening or elongation of the stem (Fig. 1B). These perturbations were all designed to take place in the 'lower' arm of the stem structure, whereas the 'upper' arm remained constant. We transfected each oligo library into ADAR1-knockout HEK293T cells (in which ADAR2 is not expressed)[50], alongside a plasmid expressing either ADAR2 or ADAR1 or neither of the two ('No-ADAR'), as a negative control. Subsequently, RNA was extracted, and the constant upper arm of each construct was reverse transcribed, PCR-amplified, and sequenced (Fig. 1C).

All B2 and mNG constructs were detected across all treatments with a mean coverage of ~4000 reads per barcode per sample across all conditions (ADAR1, ADAR2, and No-ADAR). No editing was observed in ADAR1 KO cells transfected with the No-ADAR vector, corroborating that all deamination activity is triggered by the two exogenously overexpressed ADAR enzymes (Fig. 1D). In ADAR-expressing cells, editing percentages between technical duplicates were highly reproducible (r > 0.99, P < 2.2e-16 for all treatments) (Fig. 1E & Figure S1A). In addition, editing measurements were independent of barcode identity, as was assessed by comparing editing levels at a subset of identical sequences with distinct barcodes (Fig. 1F & Figure S1B).

The editing patterns in the B2 and mNG constructs following ADAR1 overexpression were well correlated with ones observed in WT HEK293T cells in which ADAR1 is expressed at endogenous levels (Fig. 1D), suggesting that ADAR overexpression is a valid approach for interrogating the rules defining the substrate specificities of ADAR enzymes. The editing patterns following overexpression of ADAR1 and ADAR2 were substantially less correlated, in line with previous reports indicating their only partially overlapping target specificity[27,28] (Fig. 1D). We also note that ADAR2 overexpression gave rise to higher levels of editing in comparison to ADAR1, in line with previous reports[51]. Western blot analysis demonstrated elevated expression levels of ADAR2 compared to ADAR1 within our system, indicating that the differences in absolute levels may be attributed to the increased expression of ADAR2 (Figure S1C). In mNG constructs, a median of ~22 out of 44 adenosines per molecule was edited in ADAR2-overexpressing cells, in comparison to ~18 in ADAR1-overexpressing counterparts (Fig. 1G). This trend was even more pronounced, with ~3 and ~12 out of 41 edited sites per molecule in B2 constructs in ADAR1- and ADAR2-overexpressing cells, respectively.

As an additional quality control, we assessed editing levels across a series of constructs in which the double-stranded stem was randomly disrupted to varying levels. Consistent with our expectations, we found that editing by both ADAR1 and ADAR2 was continuously disrupted with progressive disruption of the secondary structure (Fig. 1H & Figure S1D). ADAR2 was slightly more resilient to the introduction of structural disruptions, consistent with previous studies showing that ADAR2 can efficiently edit shorter double-stranded substrates than ADAR1[38,52]. Collectively, these analyses establish that the two synthetic constructs and their perturbed counterparts are edited by both ADAR1 and ADAR2, yet these two enzymes are associated with both varying levels and different patterns of editing.

### ADAR2-mediated editing is induced 26 nt upstream of structural disruptions

We next sought to assess whether structural disruptions within dsRNAs induce ADAR2-mediated editing at a fixed offset, given our

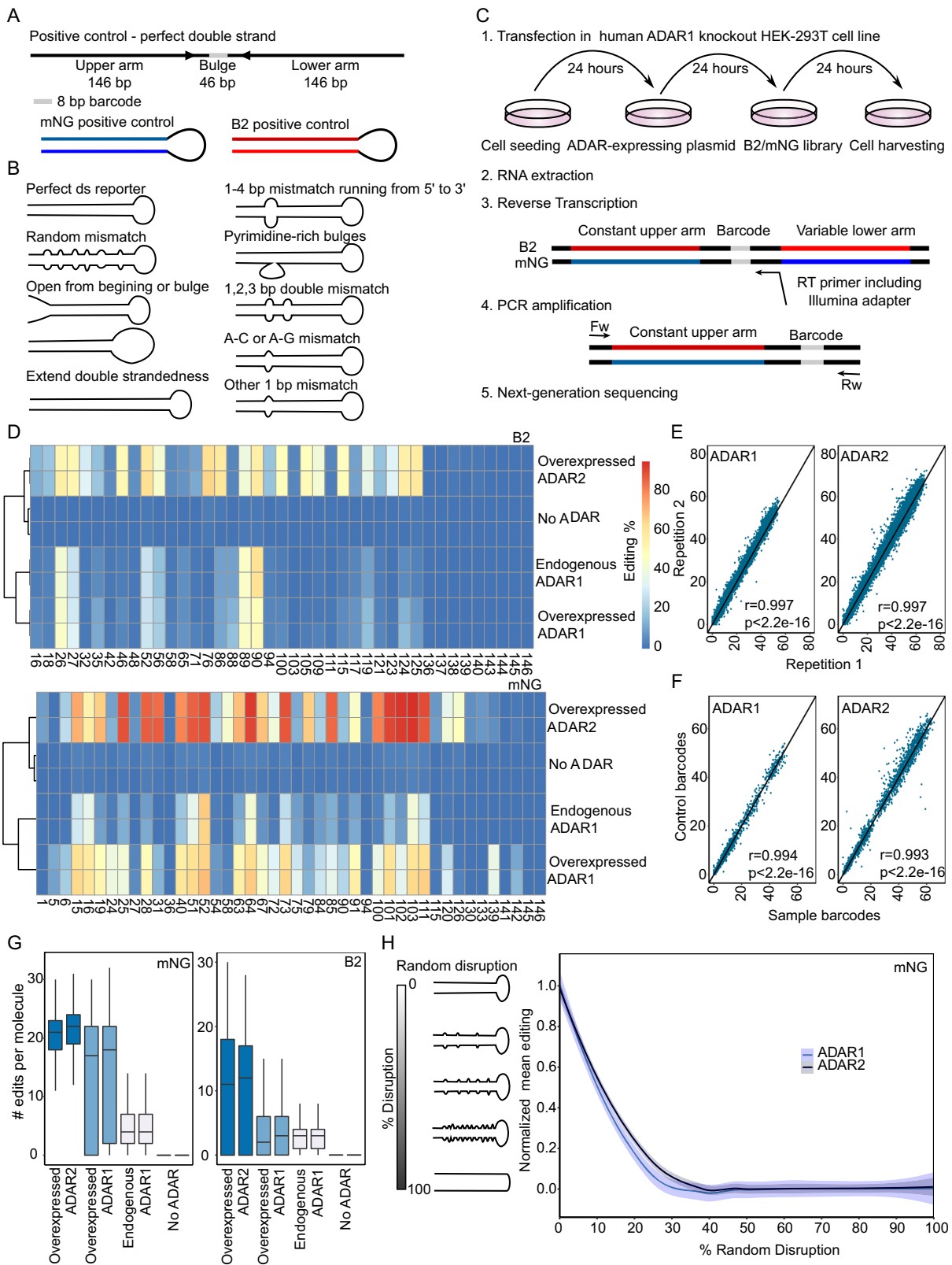

previous discoveries of a −35 bp offset for ADAR1[49]. To explore this, we analyzed the series of constructs into which we had systematically introduced secondary-structure disrupting sequences−either in the form of mismatches or of bulges−throughout the stem. Indeed, in both ADAR1- and ADAR2-overexpressing cells, increased editing levels were observed at a fixed offset (Fig. 2A–C & Figure S2A). In the case of ADAR1-overexpressing cells, we recapitulated our previous

observations of increased editing levels 35 bp upstream and 30 bp downstream of structural disruptions (Fig. 2B, C)[49]. In contrast, in ADAR2-overexpressing cells, structural disruptions led to increased editing levels in a window between 23-31 nt upstream of the structural disruption, peaking at position −26 nt (Fig. 2B, C). Though the magnitude of the increase in editing levels at position −26 following ADAR2 overexpression was lower than the increase at position −35 following

**Fig. 1 | Systematic screening of ADAR2 synthetic substrates in ADAR1 knockout HEK293T cell lines. A** Design of double-stranded reporters. B2 and mNG are based on a mouse non-coding B2 element and the mNeonGreen gene, respectively. **B** Repertoire of sequence series in B2 and mNG libraries. **C** Experimental pipeline: Expression of the synthetic libraries in ADAR1-knockout HEK293T cells, which exogenously overexpress ADAR1 or ADAR2, and library preparation. RNA was extracted, and the constant arm and barcode of each construct were reverse transcribed. Subsequently, PCR amplification and sequencing using Novaseq 6000 platform with a 300 bp kit were performed. **D** A-to-I editing levels in the B2 (upper diagram) and mNG (lower diagram) perfect double-stranded constructs in No-ADAR, ADAR1-overexpressing or ADAR2-overexpressing ADAR1-KO HEK293T cells, and wild-type HEK293T cells. **E** Correlation of A-to-I levels among technical duplicates in cells overexpressing either ADAR1 or ADAR2. Each dot depicts the editing percentage of each adenosine in each construct of the B2 oligo library. The Pearson correlation coefficient is displayed, along with the associated statistical significance

based on a two-sided test **F** Correlation of editing levels in B2 constructs that differ in the barcode sequences. Two-sided tests were based on Pearson's correlation coefficient, which follows a t-distribution with degrees of freedom 'number of observations −2' assuming normal distributions **G** Boxplots representing the distribution of numbers of editing events in the single mNG/B2 perfect double-stranded molecules in either No-ADAR cells, ADAR-overexpressing cells, or wild-type HEK293T cells. Data is visualized via box-and-whisker plots, with the central line denoting the median, box edges representing the interquartile range (from the 25th to the 75th percentile), and whiskers indicating the 1.5 times interquartile range. **H** Min-Max normalized mean editing percentage within a subset of mNG constructs featuring incremental 5% disruptions of double-strandedness. Data from both replicates are combined, and a LOESS fit (in blue) is applied to the Min-Max normalized mean editing percentage. The gray band surrounding the regression line represents the 95% confidence interval.

---

ADAR1 overexpression (-1.3–1.5 mean fold at position −26 in comparison to -3.3-6 mean fold at position −35), the phenomenon was reproducibly observed across the two different constructs as well as using different forms of structural disruption including mismatches of varying lengths (Fig. 2B & Figure S2B–G) and pyrimidine-rich bulges (Fig. 2C & Figure S3A–C). The increase in editing levels at position −35 and −26 in ADAR1- and ADAR2-overexpressing cells, respectively, was dependent on the size of the mismatch, with the highest median editing increase observed in constructs carrying 3 nucleotide mismatches (Fig. 2D). In parallel, the introduction of mismatches also led to a reproducible negative signal (indicative of adenosines resistant to editing) that was distributed in a complex—yet highly reproducible—manner with respect to the structural mismatches. The negative signal extended between positions −26 and +29. The signal was at its minimum at position 0 and +1 for ADAR1 and ADAR2, respectively, consistent with previous reports[53], with two local maximums at positions −9 and positions +6/+7 in both ADAR1 and ADAR2, and an additional ADAR2-specific local maximum at position +15 (Fig. 2B).

In the case of ADAR1, we had previously found that structural disruptions led to a symmetric induction of editing, resulting in induced editing 35 bp upstream of the structural disruption on the 'upper' arm of the dsRNA, and in parallel also resulting in induced editing 35 bp upstream of the structural disruption on the 'lower' arm[49]. Given that all results obtained thus far had only been on the basis of sequencing of the 'upper' (and invariable) arm, we next amplified and sequenced the 'lower' variable arm of each B2 construct from ADAR1-KO HEK293T cells overexpressing ADAR2 (Fig. 2E). A prominent peak 26 bp upstream from the structural disruption was observed on the opposite strand (Fig. 2F), indicating that the induction of editing by human ADAR2 is symmetric and orientation-dependent at a fixed interval as was the case for ADAR1, but in this case 26 nt upstream from structural disruptions.

### Differences in editing offsets among ADARs are mediated by double-stranded RNA binding domains

We next sought to understand why structural disruptions led to editing at an offset of 35 nt in the case of ADAR1, but of 26 nt in the case of ADAR2. To explore whether the offset was dictated by the catalytic domain of the two ADAR enzymes or by the RBDs, we designed two ADAR variants, by swapping the RBD domains among the ADARs: (1) An 'ADAR2-RBDs_ADAR1-deaminase' variant, harboring the catalytic domain of ADAR1 fused to the two RBDs originating from ADAR2, and (2) An 'ADAR1-RBDs_ADAR2-deaminase' variant, harboring the catalytic domain of ADAR2 fused to the three RBDs originating from ADAR1. We next used the above-described human ADAR1-depleted system, into which we transfected B2 and mNG oligo libraries along with plasmids overexpressing these two ADAR

variants. The two hybrids gave rise to deamination activity on both B2 and mNG positive control constructs (Figure S4A, B), albeit at substantially reduced levels in comparison to the WT counterparts (Fig. 3A). Remarkably, we found that the offset size segregated with RBDs: 'ADAR1-RBDs_ADAR2-deaminase' showed induced editing levels at position −35, recapitulating the patterns observed in WT ADAR1 expressing cells. In parallel, 'ADAR2-RBDs_ADAR1-deaminase' exhibited induced activity at roughly −30 nt, as had similarly been observed for ADAR2 (Fig. 3B, C; Figure S5A, B; Figure S5E, F). Thus, these findings suggest that the size of the offset is encoded by the differential RBD architecture.

How do the different RBDs give rise to differential offsets? We hypothesized that the RBDs might serve as molecular rulers and that the size of the offset might scale roughly linearly with the number of RBDs. Under this scenario the offset in ADAR2 with respect to ADAR1 might reflect the loss of one RBD in ADAR2, harboring 2 RBDs, in comparison to ADAR1, harboring 3 RBDs. To test this hypothesis, we designed two ADAR2 variants harboring only a single RBD by either maintaining only the first or only the second RBD, with the anticipation that these might lead to an offset potentially even smaller than −26 nt. Both mutants were active within cells, albeit at drastically different levels (Figs. 3A, S4A, S4B), with the mutant harboring only the first RBD exhibiting very low levels of activity, in contrast to the RBD2-harboring mutant that gave rise to higher levels of editing than WT ADAR2, consistent with[54]. Nonetheless, in both cases, the size of the offset remained fixed at roughly −26, similar to WT ADAR2 (Fig. 3D; Figure S5C, S5G). These findings thus suggest that the size of the offset is not determined by the number of RBDs.

The above results left open the possibility that the effect of the number of RBDs might be threshold-dependent. Under such a scenario, one or two RBDs might invariably give rise to an offset of −26, but never to an offset of −35, for which a third RBD would be required. To test this possibility, we selected two additional ADAR homologs from *Suricata suricatta* and *Octopus vulgaris*, harboring one and two RBDs, respectively, to assess whether these invariably gave rise to editing at an offset of −26 nt. The two ADAR enzymes elicited deamination activity on B2 or mNG positive control constructs (Figure S4A, B), albeit at varying levels and with differences in substrate selectivity (Fig. 3A). Interestingly, the two ADAR1 homologs gave rise to different offsets: suricata ADAR gave rise to an offset of −35 similar to human ADAR1, whereas octopus ADAR displayed a peak at position −28, similar to human ADAR2 (Fig. 3E; Figure S5D, H). Collectively, these findings thus establish that while the size of the offsets is encoded within the RBD architecture, it is not encoded in the number of RBD domains either in a linear or a threshold-dependent manner, and instead it appears to be an inherent property that can be encoded even within a single RBD (see Discussion).

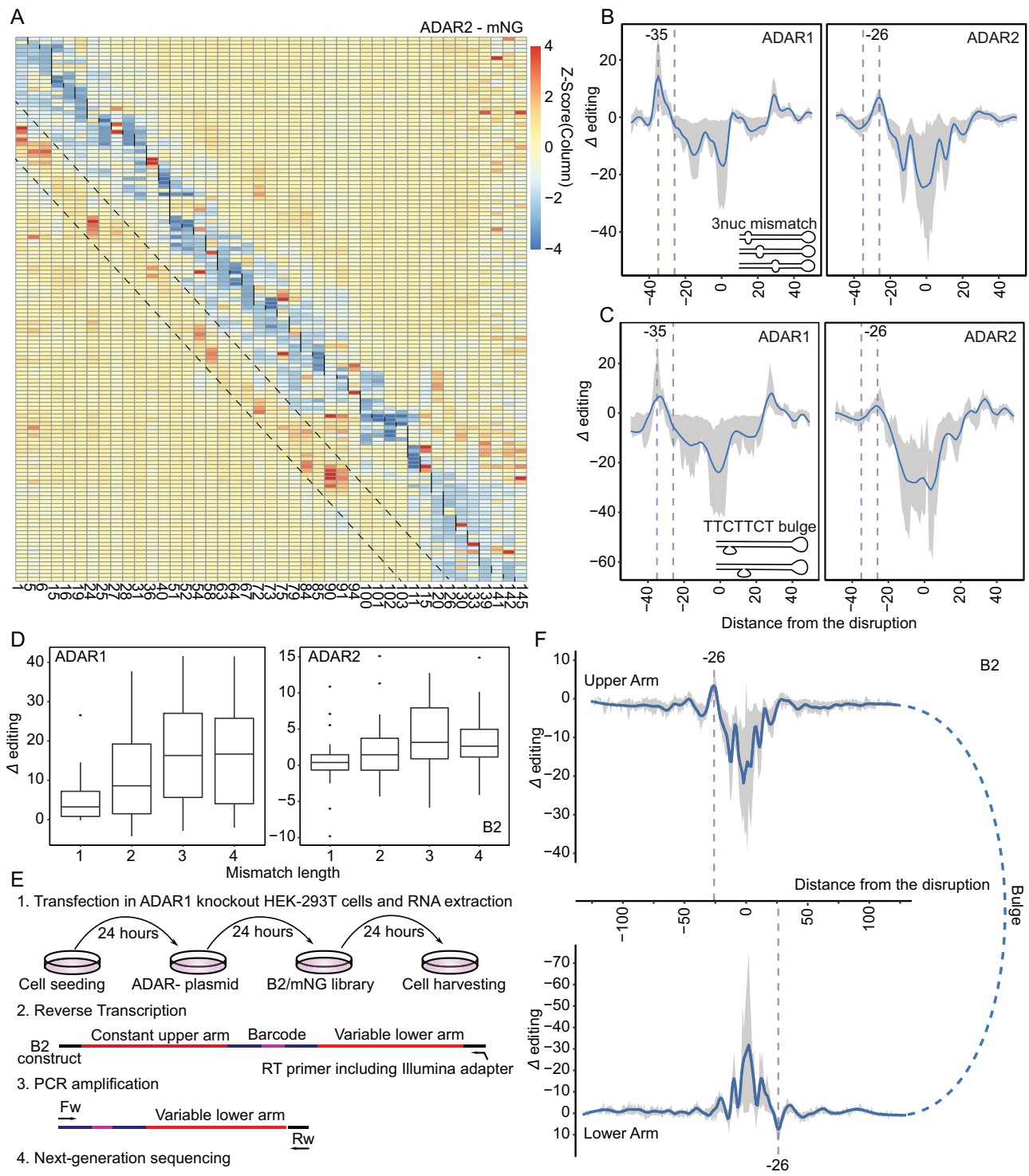

## Evidence for offset-mediated editing is observed in endogenous targets in human tissues

Our conclusions thus far relied on offsets observed on synthetic constructs in human cell lines, leaving open the question of whether these findings are of relevance to editing at endogenous targets in human tissues. To explore this question, we set out to assemble a dataset of all endogenously edited sites within transposable elements harboring a resolvable secondary structure. Given the uncertainty, in most cases, regarding which elements base pair with each other, we limited our analysis to a subset of 624 edited transposable elements stringently selected based on having only a single transposable element of opposite orientation in close proximity. We then predicted the secondary

structures of each of these elements, and retrieved editing levels for each adenosine forming part of the structures. Editing levels were calculated based on surveying the entirety of the GTEX dataset (spanning 9125 samples from 47 tissues, collected from 548 donors). On the basis of this collection, we next annotated each of the 40 nt flanking each editing site as being either closed (if it resided in a stem) or open (if it resided in a loop or in a bulge). Finally, we generated an editing metaplot, wherein we calculated the extent to which structuredness within a window of 80-nt centered around an editing site impacted editing levels. Remarkably, this analysis revealed two regions in which an open structure led to increased editing: A major window 35 nt downstream of the edited site and a more minor window 30 nt upstream of the edited

**Fig. 2 | ADAR2-mediated editing is induced at a constant interval of 26 bp upstream from structural disruptions. A** Heatmap of a 3-nucleotide mismatch running from 5' to 3' throughout the double-stranded RNA. Rows represent structurally disrupted constructs at specific positions, columns represent adenosine positions, and Δ editing is color-coded after Z-score transformation (mNG series). Vertical black lines mark the 3 nt mismatch location, and dashed lines highlight ADAR2-mediated editing increase upstream. **B** ADAR1- and ADAR2-mediated editing offsets based on the subset of 3-nucleotide mismatch running throughout the mNG and B2 sequences. Mismatches differentially located in each construct get centered at 0 on the x-axis. The Δ editing level on the y-axis represents the change of the editing level of an adenosine, normalized to the perfect double-stranded construct. Fitted curves depict the *LOESS* fit (blue-colored) of Δ editing with a span of 0.05 and the gray-shaded region spans the 25th percentile and 75th percentile values of Δ editing per distance. Only adenosine positions, which have greater than 1% in editing on the perfect double-stranded construct, were considered. Vertical dashed lines are placed at −26 and −35. **C** Subset of TTCTTCT bulges running throughout the mNG and B2 sequences. Data is shown as the LOESS fit curve (blue-colored) of Δ editing with a span of 0.11 and the gray-shaded region spans the 25th percentile and 75th percentile. **D** The mismatch size affects ADAR1- and ADAR2-mediated editing on adenosines located at −35 and −26 downstream from the mismatch, respectively. Data is visualized via box-and-whisker plots, with the central line denoting the median, box edges representing the interquartile range (from the 25th to the 75th percentile), and whiskers indicating the 1.5 times interquartile range. **E** Library preparation: RNA was extracted, the B2 variable lower arm and barcode were reverse transcribed, and subsequently PCR amplification and sequencing using Novaseq 6000 platform with a 300 bp kit were performed. **F** Depiction of the subset of 3-nucleotide mismatch running throughout the stem (B2) in ADAR1-knockout HEK293T cells overexpressing ADAR2. Constant and variable arms are illustrated under each other, and nucleotide locations are aligned. Data is shown as Fig. 2B.

site (Figure S6A). This analysis is thus fully consistent with our analyses on ADAR1, and demonstrates that the −35/30 nt rule is not only discernible in synthetic constructs in cell lines, but also apparent in an unbiased analysis in tissue-derived RNA. We furthermore found these results to be reproducible also when confining them to specific tissues (not shown), again pointing at their generality.

In the above analysis, we did not observe evidence for offset-mediated editing also at position −26, which is consistent with the fact that most targets within transposable elements are mediated by ADAR1, rather than by ADAR2. Drastically fewer editing sites dependent exclusively on ADAR2 were characterized, and thus it was not possible to conduct the above analysis for ADAR2-associated editing sites. To assess whether we could find evidence for offset-mediated editing of ADAR2, we considered a set of 10 sequences which harbor highly curated well-established targets of ADAR2[28]. Among those sequences prediction of secondary structure could be established with high confidence for 7 sequences. An examination of ADAR2-mediated editing sites within these targets revealed that in 5 of the 7 most structures (in FLNA, GRM4, FLNB, and NOVA1), the highly edited target harbored a mismatch 26 nt away (Fig. S6B). Moreover, in the cases of GRIA4, a mismatch was present at position +27, whereas in SON it was present at position +25. While this evidence is more anecdotal in nature, it again points to the generality of our results and their relevance to editing at endogenous targets.

## 26-bp offset rule can improve the efficiency of ADAR2-mediated targeted editing

To explore whether the recently identified −26 nt rule of ADAR2 might lend itself to improved design of ADAR recruiting therapeutics, we designed ADAR-recruiting RNAs (arRNA) to elicit editing on five distinct targets harboring distinct consensus motifs located on four different endogenous transcripts: PPIB-ORF:UAG, GAPDH-UTR:UAG, SMAD4-UTR:CAG, PPIB-UTR:UAG and STAT1-ORF:UAU via recruitment of exogenously expressed ADAR2. For each of these targets, we designed three arRNA constructs: (1) a 151-nt long arRNA containing a C opposite to the target A located between two 75-nt stretches which are perfectly complementary to the endogenous transcript. Such constructs were used in[23,55] and served as a positive control; (2) an arRNA as in (1) but harboring 3-bp mismatch 26 or 27 nt upstream from the target adenosine, and (3) an empty vector serving as a negative control (Fig. 4A). Consistent with our expectations, we found that in 2 of the 5 cases (GAPDH-UTR and SMAD4-UTR), the 3-nt disruptions significantly increased editing levels with respect to the positive controls, and in a third case (PPIB-ORF) the same trend was observed albeit it did not pass statistical significance (Fig. 4B). The relatively low increase in these cases as well as the absence of an increase in the two remaining cases are consistent with the relatively mild effect size of induced editing at position −26 (Fig. 3B) and may be suggestive of context-specificity remaining to be uncovered.

In some clinical contexts, it could potentially be beneficial to induce editing only in cells expressing one of the two ADAR enzymes, for instance in order to achieve selectivity in brain tissues in which ADAR2 is expressed. Given the different offsets at which ADAR1 and ADAR2 induce editing, we sought to assess whether this could be leveraged to achieve such selective editing. Indeed, we found that an arRNA with a structural disruption at an offset of 35 nt selectively induced editing by ADAR1, and not by ADAR2, in comparison to a positive control lacking a structural disruption (Fig. 4C, D). Conversely, an arRNA with a structural disruption at a 26 nt offset selectively induced editing by ADAR2, and not by ADAR1 (Fig. 4D). These results thus suggest that engineered structural disruptions at fixed offsets can be utilized to tune the relative susceptibility of targets to editing via ADAR1 vs ADAR2.

Finally, we sought to assess whether the introduction of structural disruptions at a 26 bp offset would not only increase on-target editing levels but also decrease off-target levels. To assess this, we amplicon-sequenced the GAPDH amplicon following targeted editing via either the 'positive control' or the 'Mismatch 27' arRNA. In this analysis, we only identified a single adenosine that was edited at levels exceeding 2% across either of these two samples. Remarkably, this position was edited at levels of 6.13% in the positive control sample, which decreased to 1.05% in the 'Mismatch 27' samples (Fig. 4E, F). This off-target site resided 26 nt downstream of the targeted adenosine, and therefore the reduced editing levels in the 'Mismatch 27' sample are likely a direct consequence of this position no longer being base-paired in the 'Mismatch 27' arRNA. With the caveat of only relying on a single off-target site, these findings suggest that rationally designed structural disruptions within arRNAs can be designed to both increase on-target rates and decrease off-target ones.

Overall, these findings lend support to the observations that structural disruptions lead to increased ADAR2-mediated editing at a fixed offset and provide a proof of principle that this rule can allow improved recruitment of ADAR2 towards target adenosines in therapeutic settings.

## Characterization of ADAR1 and ADAR2 sequence selectivity across diverse ADAR variants

The experimental design of the oligo-array libraries employed in this study had been primarily geared towards interrogating the impact of RNA secondary structure on editing. Nonetheless, the availability of measurements of editing levels across distinct sites and in varying sequence contexts allowed investigating the impact of sequence on editing, and the extent to which this varied across the eight ADAR variants interrogated here.

We found that across all ADAR enzymes, the position immediately upstream of the edited site was depleted of G at the upstream position, consistent with[38,39,56]. The position immediately downstream displayed less of a bias, consistent with[35,38,39,57]. (Fig. 5A). We next explored the

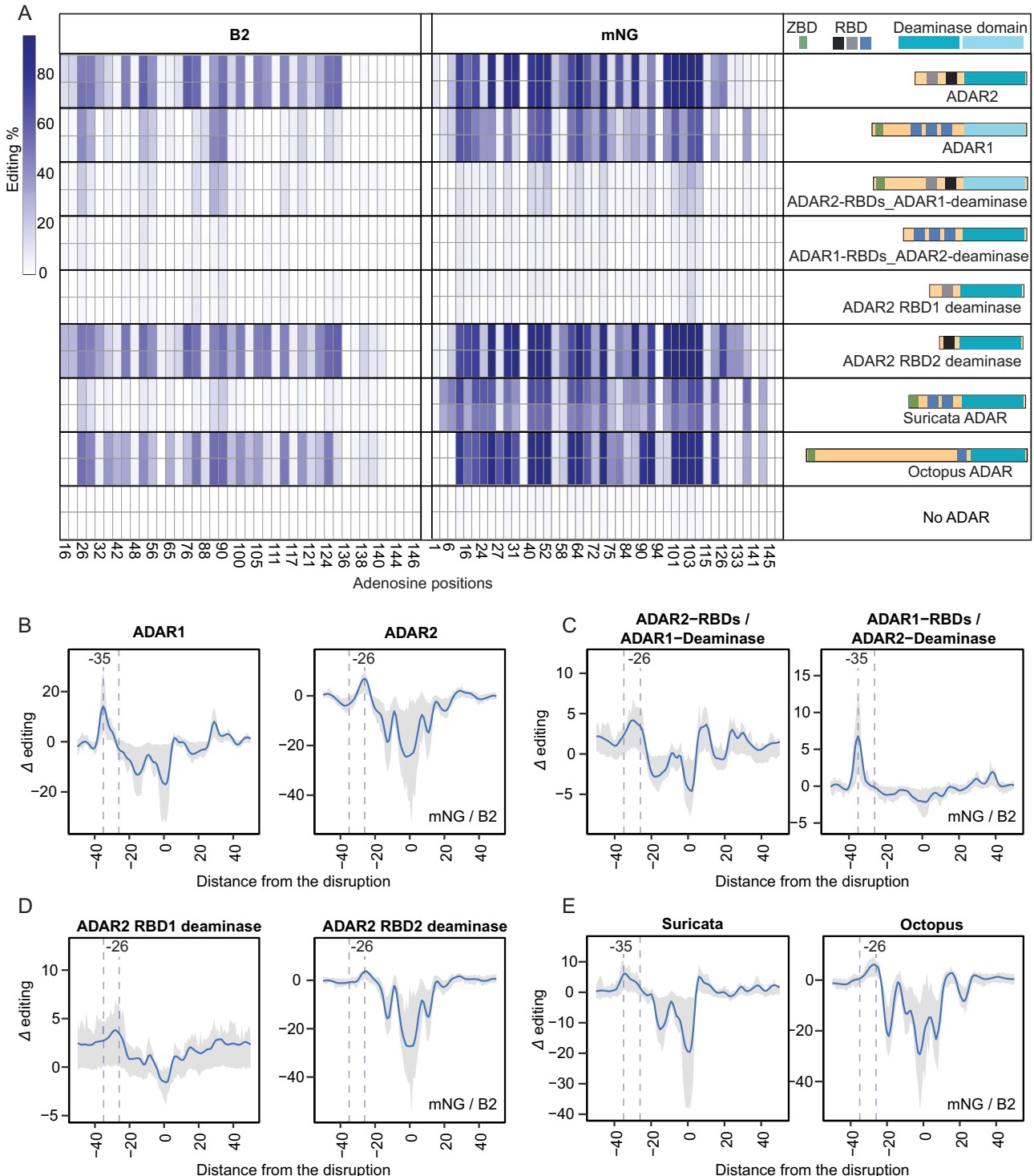

**Fig. 3 | Editing is induced at ADAR-dependent fixed intervals upstream from structural disruptions. A** Heatmap of A-to-I editing levels in the perfect double-stranded constructs in No-ADAR and ADAR-overexpressing ADAR1-KO HEK293T cells. The adenosine positions of the B2 and mNG perfect double-stranded constructs are depicted at the bottom of the heatmap. The illustrations of each ADAR including the RBDs and deaminase domain are depicted on the right side. ZBD: Z-binding domain; RBD: RNA binding domain. **B** ADAR1- and ADAR2-mediated editing offset based on subsets of 3-nucleotide mismatch running throughout the mNG and B2 sequences. Data is shown as the LOESS fit curve (blue-colored) of ∆ editing with a span of 0.05, and the gray-shaded region spans the 25th percentile and 75th percentile values of ∆ editing per distance. **C** Editing offset based on subsets of 3-nucleotide mismatch running throughout the mNG and B2 sequences in 'ADAR2-RBDs_ADAR1-deaminase'- and 'ADAR1-RBDs_ADAR2-deaminase'-overexpressing cells. Data is shown as Fig. 3B. **D** Editing offset retrieved from subsets of 3-nucleotide mismatch running throughout the mNG and B2 sequences in 'ADAR2-RBD1 deaminase'- and 'ADAR2-RBD2 deaminase'-overexpressing cells. Data is shown as Fig. 3B. **E** Editing offset based on subsets of 3-nucleotide mismatch running throughout the mNG and B2 sequences in 'Suricata ADAR'- and 'Octopus ADAR'- overexpressing cells. Data is shown as Fig. 3B.

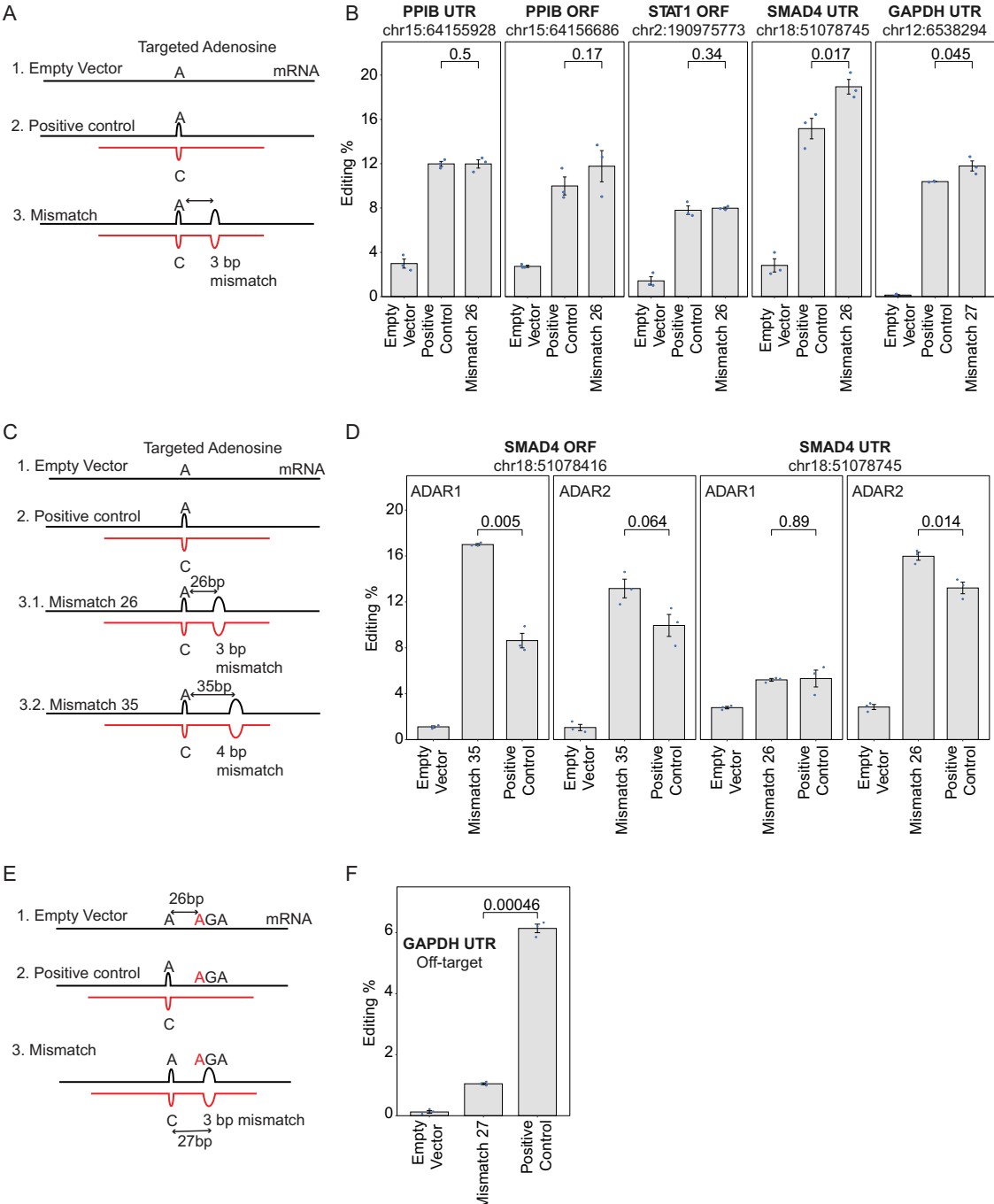

**Fig. 4 | Targeted editing of endogenous transcripts using engineered ADAR2- and ADAR1-recruiting RNAs. A** Scheme of arRNAs targeting endogenous transcripts of *PPIB*, *SMAD4*, STAT1, and GAPDH. (1) The empty vector has no targeting oligo. (2) Positive control construct is a 151-bp-long complementary oligo, with a T to C mismatch opposite of the targeted A. (3) Mismatch 26 construct consists of an arRNA as in (2) but including a 3-bp mismatch at 26 or 27 bases away from the target A site. **B** Quantification results showing the editing levels on targeted adenosine of the *PPIB*, *SMAD4*, STAT1, and GAPDH transcripts in ADAR2-expressing cells. Data is shown as the mean ± s.e.m.(standard error of the mean) with *n* = 3 independent experiments. The pairwise comparisons were evaluated using a one-tailed t-test and the corresponding *p*-values are shown on the top of the barplots. **C** Scheme of arRNAs targeting endogenous transcripts of *SMAD4*. (1) The empty vector has no targeting oligo. (2) Positive control construct is a 151-bp-long complementary oligo, with a T to C mismatch opposite of the targeted A. (3.1) Mismatch 26 construct

consists of an arRNA as in (2) but including a 3-bp mismatch at 26 bases away from the target A site. (3.2) Mismatch 35 construct consists of an arRNA as in (2) but including a 4-bp mismatch at 35 bases away from the target A site. **D** Quantification results showing the editing levels on targeted adenosine of the *SMAD4* transcript in ADAR1- and ADAR2-expressing cells. Data is shown as the mean ± s.e.m.(standard error of the mean) with *n* = 3 independent experiments. The pairwise comparisons were evaluated using a two-tailed t-test and the corresponding *p*-values are shown on the top of the barplot. **E** Scheme of arRNAs targeting endogenous GAPDH transcript. **F** Quantification results showing the editing levels on off-targeted adenosine of the *GAPDH* transcript in ADAR2-expressing cells. Data is shown as the mean ± s.e.m. (standard error of the mean) with *n* = 3 independent experiments. The pairwise comparisons were evaluated using a two-tailed t-test and the corresponding *p*-values are shown on the top of the barplot.

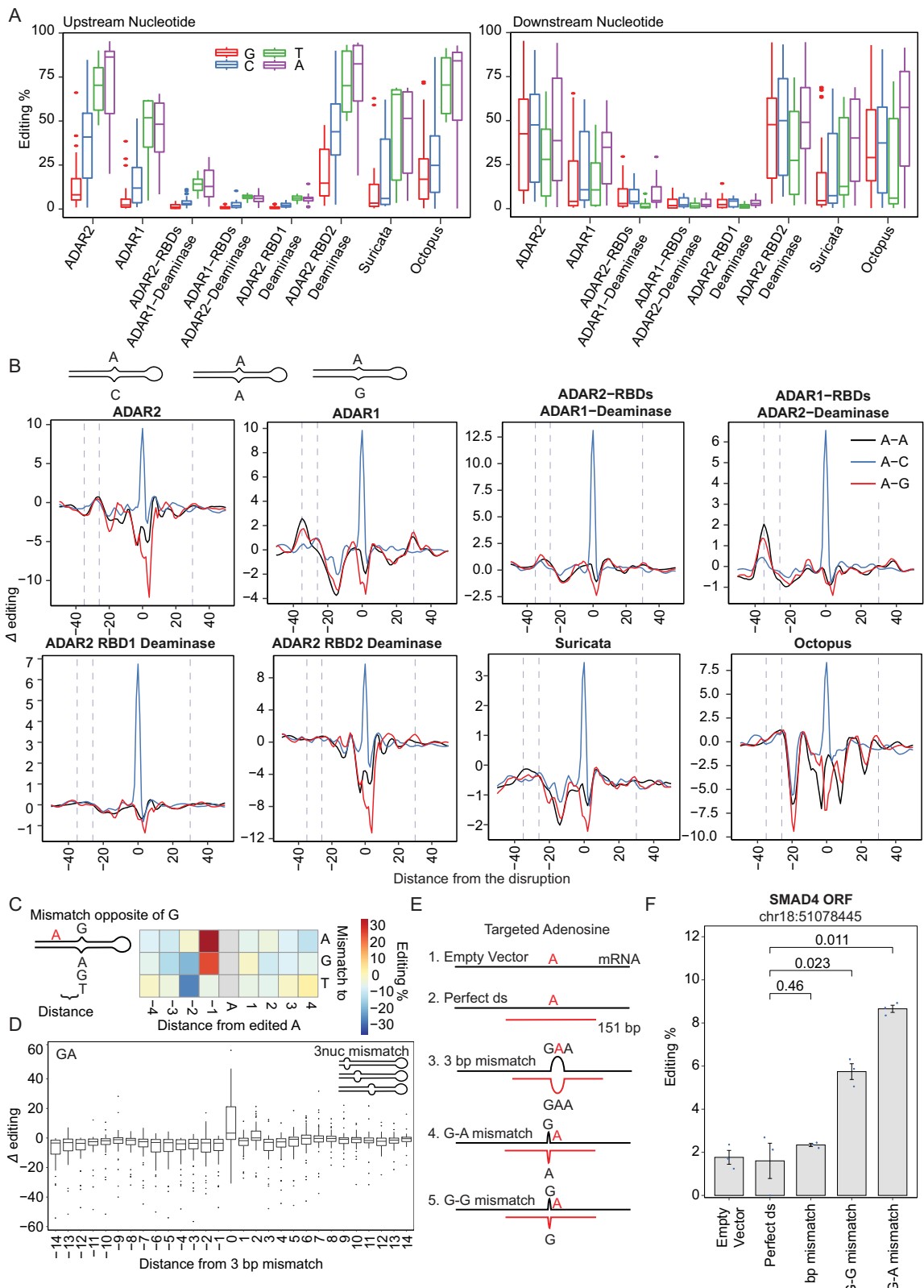

extent to which the identity of nucleotides opposite of the target adenosine impacted editing across the different ADAR variants. We found that editing by all ADAR variants was induced when a C was introduced opposite of the target A (Fig. 5B). Introduction of A-A or A-G mismatches opposite of the edited site both substantially decreased editing at the targeted position and gave rise to increased editing at a −26 bp offset in ADAR2-expressing cells (Fig. 5B). Finally,

we extended this analysis to mismatches occurring in the vicinity of the edited site. This analysis revealed that editing at adenosines in a GA context (underlined A is edited) tends to be substantially higher when the nucleotide opposite of the G at position −1 is a guanosine, and even more so an adenosine (Fig. 5C). We further found induced levels of editing when a 3-nt mismatch was centered around an edited site in a 'GA' context (Fig. 5D). These findings were consistently observed

**Fig. 5 | Sequence and structure elements involved in editing nucleation and termination among ADARs. A** Upstream (left) and downstream (right) nucleotide preference in ADAR-specific editing. Editing levels correspond to adenosines along both double-stranded constructs, but As near the loop were excluded. Data is visualized using box-and-whisker plots, where the central line represents the median, the box edges depict the interquartile range, and the whiskers show 1.5 times the interquartile range. **B** Constructs characterized by a systematic C, A or G base opposite to A along the stem. Line charts show the effect of the different mismatches on editing. The *Δ* editing level on the y-axis represents the change in the editing level of an adenosine, normalized to the double-stranded construct. Fitted curves depict the *LOESS* fit of *Δ* editing with a span of 0.07. **C** Subset of G-mismatching bases that neighbor the edited sites. Left—Graphical scheme. Right —On the heatmap, the x-axis shows the distance of the disruptions to the A site while the y-axis represents the base to which a G is opposite. **D** Effect of 3-nt

mismatch running through the stem on adenosine within the GA sequence context in ADAR2-expressing cells. Mismatches differentially located in each construct get centered at 0 on the x-axis. The *Δ* editing on the y-axis represents the change of the editing of an adenosine, normalized to the perfect double-stranded construct. The box plot depicts the distribution of *Δ* editing levels per distance. Data is shown as Fig. 5A. **E** Scheme of arRNAs targeting endogenous *SMAD4* transcript. (1) The 'Empty Vector' as negative control. (2) 'Perfect ds' construct is a 151-bp oligo complementary to the transcript. (3) '3 bp mismatch' construct consists of an arRNA as in (2) but containing a 3-nucleotide mismatch opposite to the target A site. (4-5) 'G-G and G-A mismatch' constructs consist of an arRNA as in (2) but including a G-G and G-A mismatches one nucleotide upstream from the A site. **F** Quantification showing the editing levels on the adenosine of the *SMAD4* in ADAR2-expressing cells, presented as mean ± s.e.m. (*n* = 3 independent experiments) and evaluated with a two-tailed t-test.

across all ADAR variants (Figure S7A, B). The facts that these sequence preferences are independent of the RBD domain structure and that they occur at sites that are in physical interaction with the deaminase domain suggest that these sequence preferences are an inherent property of the ADAR deaminase domains and that they are shared across ADAR1 and ADAR2 homologs.

Given that adenosines in GA contexts are typically edited at low efficiencies, we sought to investigate whether editing in GA contexts could be induced via the introduction of arRNAs designed to harbor a G-G or a G-A mismatch at position −1, or via guides introducing a 3-nt mismatch at the edited site. Indeed, we found that an arRNA harboring a G-A mismatch yielded the highest editing level within a SMAD4-ORF target, followed by arRNA harboring a G-G mismatch, whereas the fully complementary arRNA yielded background levels of editing (Fig. 5E, F). These findings are in line with reports by[58,59]. In parallel, we also found that introduction of such GA mismatches, while increasing on-target effects, was also associated with increased off-target effects at an offset of 26 nt (Figure S7C). Collectively, our findings establish how editing at target sites can be induced either by introducing mismatches at a relatively distant fixed offset via a mechanism impacting recognition through the RBDs, or in close vicinity to the target site via a mechanism likely impacting recognition through the deaminase domain.

## Discussion

Despite widespread interest in unraveling the determinants guiding the selectivity of ADAR1 and ADAR2, these have remained poorly understood and to a considerable extent unpredictable. It has been previously suggested that the basis for selectivity resides within mismatches[37], bulges, loops[53], and long-range tertiary pseudoknots[60,61]. Such structural elements are evolutionarily conserved[60,62] suggesting that the secondary[35] and tertiary RNA structures[63] play an important role in regulating the editing efficiency and specificity. Accordingly, mismatches and bulges have also been included in the design of prior arRNA recruiting modalities[22,55]. Yet, the rules governing such selectivity—e.g., where do structural mismatches contribute to editing? When are they prohibitive?—have remained poorly understood. Our study contributes two key insights to our understanding: First, we establish a simple rule, namely that structural disruptions of diverse types (bulges, mismatches) will give rise to induced ADAR2-mediated editing at a fixed offset of 26 bp upstream of the disruption, contrasting with ADAR1 which induces editing at a 35 bp offset. Second, we uncover that these distinct offsets by the two ADARs are encoded via the distinct RBD domains of the two enzymes. Importantly, our analyses of GTEX-based data confirm the relevance of offset-mediated editing to endogenous sites within human tissues.

Our work uncovers interesting commonalities and differences between the two ADAR enzymes. Activity by both enzymes is induced at a fixed offset from structural disruptions. In both cases, there is substantial evidence for symmetricity, as is evident from comparing

the upper and lower strand editing levels. Moreover, in both cases, the induction of editing is orientation-specific, with editing being induced on both strands upstream of the structural disruption. However, the size of the offset is different (−35 vs −26 nt). In addition, the magnitude of induction is also different, with more dramatic effects being typically observed for ADAR1 than for ADAR2. Finally, for ADAR1 in addition to the major peak at −35, we had also observed a more minor peak in editing activity 30 bp downstream of the edited site. We do not observe such a downstream peak for ADAR2. This may either reflect a difference in the mechanism driving induced editing, or the lower dynamic ranges which may limit us from clearly observing such a secondary, more minor peak for ADAR2.

A major question left open by our study is the basis for the different offsets of ADAR1 and ADAR2. While based on the RBD swapping experiments it is clearly encoded by the RBD architecture, we rule out that this is a function of the number of RBDs, as offsets of 26 and 35 nt are achieved by variants and mutants with a distinct number of RBDs. Another possibility is that the difference in offset is not due to the difference in domains, but to the difference in the size of the linker between the RBD and the deaminase domain. However, we can largely rule out this possibility as well, because in our RBD swapping experiments between ADAR1 and ADAR2 we had maintained the original linkers, and the offset sizes segregated with the RBDs. Given that single amino acids in the RBD were shown to be important in RNA recognition and binding[43], it is possible that the basis for the difference in selectivity between the two enzymes lies within such individual changes. Dissecting this systematically via genetic approaches is rendered challenging, given that mutations within RBDs oftentimes also abolish editing. Indeed, six additional RBD-disrupting ADAR mutants that we generated over the course of this study (data not shown) failed to show any substantial editing activity, consistent also with previous observations[64]. Our inability to obtain active mutants harboring only the catalytic domains and none of the dsRBDs (data not shown) also prevents us from formally ruling out the possibility that part of the offset might be directly mediated by the dsRBD itself. We anticipate that the structural dissection of these two enzymes bound to RNA targets will contribute towards answering this question.

In attempting to understand the basis for a 26 nt offset of ADAR2, we found two potentially relevant clues in the literature. First, in a structural study of the Glu receptor target in complex with the ADAR2 RBDs, each of the two domains was found to associate with 12-14 nt. Thus, 26 nt is well within the range of the size that would be protected by two RBD[43]. While our findings suggest that a 26 nt offset can also be maintained via ADAR mutants and variants harboring a single RBD, they do leave open the possibility that an offset of 26 nt could be the combined outcome of the RBDs of two ADAR enzymes acting as a dimer, given that both ADAR1 and ADAR2 act as homodimers[65–68]. Second, our studies resonate to some extent with findings that ADAR substrates are distributed periodically at ~50 bp intervals from each other[69]. Given that we find editing induced 26 nt upstream of structural

disruptions on the top strand, but also at 26 nt upstream of the disruption on the lower strand (Fig. 2F), and given our previous observations on editing symmetricity[49], it is tempting to speculate that structural disruptions could serve as a mechanism spacing edited sites at ~52 bp intervals from each other. However, in the cited study[69] the same intervals were observed for ADAR1 and ADAR2, whereas different intervals would be predicted for ADAR1 vs ADAR2 based on such a model and our findings, and thus it is unclear to us whether these findings are mechanistically related.

In our study, we also perform proof-of-principle experiments demonstrating that our improved understanding of editing specificity by the two ADAR enzymes lends itself towards the improved design of ADAR recruiting RNA sequences. We demonstrate that the offsets at a fixed distance can enhance on-target editing levels at the specified targets, potentially reduce off-target editing, and can provide some level of control over which of the two enzymes mediates it. While the effect sizes obtained in our hands are in most cases relatively modest, we anticipate that they might potentially be boosted, if combined with more potent arRNAs, such as chemically modified ones[22].

Our study also suffers from several limitations. First, it was primarily geared to dissect *relative* differences in substrate selectivity between ADAR1 and ADAR2, rather than *absolute* differences. Given that it relies on overexpression of the two enzymes, and to unequal levels, such absolute differences—involving comparison of the same target across experiments, rather than of different targets within a single sample—need to be interpreted with caution. Second, our study was conducted using a synthetic set of sequences within a human cell line. While this raises questions as to its relevance on endogenous substrates in human cells and tissues, our ability to confirm offset-mediated editing in GTEX cell lines strongly supports the relevance of our conclusions. Third, our study was primarily designed to monitor the impact of structural perturbations on editing. While we do also utilize our measurements to interrogate the role played by sequence (Fig. 5), the sequence space that we sample is limited, which may potentially skew the results. While our results in this section generally are consistent with prior literature, we do not observe a clear enrichment for a G downstream of the edited site (Fig. 5A), in contrast to previous studies[35,38,39]. This may reflect the limited, monitored sequence contexts in our study- only 29 AG dinucleotides were monitored, 11 of which (38%) in a 'GAG' context, whereby it is known (and also observed here) that an upstream G plays an inhibitory role on editing. Finally, given that editing is a product of both sequence and structural preferences, it will be important to simultaneously perturb both, in order to understand to what extent these two dimensions act in an additive or synergistic manner. This dimension could not be profiled using our current design.

Collectively, our findings shed light on the mechanisms underlying the only partially overlapping target spectrum of ADAR1 and ADAR2, while advancing our technical toolkit to target these two enzymes towards clinically relevant targets.

## Methods

### ADAR plasmid generation

Full-length human ADAR2 (UniProt: P78563-2), ADAR2 RBD1 Deaminase, and ADAR2 RBD2 Deaminase coding sequences were amplified from the AAVS1-hADAR2, pYES-DEST52-hADAR2-dRBM1-Deaminase domain, and pYES-DEST52-hADAR2-dRBM2-Deaminase domain plasmids, respectively, using primers that included XbaI and EcoRI sites. Full-length human ADAR1 (UniProt: P55265-5) was amplified from the AAVS1-hADAR1 plasmid by primers containing XbaI and HindIII sites. All of those PCR products (primers in Supplementary Data 1) were subsequently digested and ligated into the corresponding restriction sites of the digested pcDNA3.1(-) vector. Additionally, both recombinant ADAR constructs contain a FLAG tag (peptide sequence DYKDDDDK) at their N-terminus.

For designing ADAR1-ADAR2 hybrid plasmids, the ADAR1-pcDNA3.1(-) and ADAR2-pcDNA3.1(-) plasmids were used as templates for PCR reactions (primers in Supplementary Data 1) using Phusion® Hot Start II DNA polymerase (Thermo Fisher Scientific). The gel-purified DNA fragments were assembled according to Gibson Assembly® Master Mix (NEB). The assembled products were transformed using a Gibson Assembly Cloning Kit (NEB), and all constructs were confirmed via Sanger sequencing on PCR-based positive clones (primers in Supplementary Data 1). Final ADAR-plasmid-containing clones were grown in ampicillin-supplemented LB liquid media, and DNA was extracted according to the QIAprep Spin Miniprep Kit (QIAGEN).

The pTwist CMV vectors containing the human-codon optimized sequences of ADAR from *Octopus vulgaris* (UniProt: A0A6P7SC-W6_OCTVU), and *Suricata suricatta* (UniProt: A0A673T544_SURSU) were ordered from Twist Bioscience. Bacteria from glycerol stocks were inoculated and grown in ampicillin-supplemented LB liquid media, and plasmid DNA was extracted as previously mentioned.

### Transient transfections

ADAR1-knockout HEK293T cells[70], which were obtained as gift from lab of Prof. Dan Stetson, were grown (37 °C, 5% $CO_2$) in Gibco Dulbecco's Modified Eagle Medium (DMEM) supplemented with 10% fetal bovine serum, 1% Penicillin and Streptomycin, and 4ug/ml Puromycin. $5 \times 10^5$ cells were plated on a 6-well plate so that cells reached 70–90% confluency at the time of the second transfection. 24 and 48 hours after cell seeding, 1.6 µg of ADAR-expressing pcDNA3.1(-) plasmid and 4 µg B2 or mNG library DNA were transfected respectively according to Lipofectamine® 2000 DNA Transfection Reagent Protocol (Thermo Fisher Scientific). 24 hours later, cell harvesting was performed.

### RNA processing and library preparation

Total RNA was extracted using Nucleozol (Macherey-Nagel), poly-A selected using oligo dT-beads (Dynabeads mRNA DIRECT Kit life tech), and DNase treated (Thermo Fisher Scientific). The upper constant arms or lower variable arms including 8 nucleotide barcodes of constructs were reverse transcribed, PCR amplified (primers in Supplementary Data 1), and sequenced using NovaSeq 6000 SP Reagent Kit v1.5 (300 cycles).

### Data analysis of NGS data

Fastq files were assessed by a custom R script (Supplementary Data 2). The read-filtering process removed reads containing wrong start and end, lacking the established barcodes, and misaligning at adenosine positions. Read 1 and 2 were merged into a single sequence by custom truncation and matching. For each barcode, the editing percentage was quantified as $(G/(A + G))*100$ at each adenosine position. $\Delta$ editing was calculated as the difference of editing levels at adenosine positions between each structurally altered sequence and perfect-double stranded construct, respectively.

### Western blotting

Lysates were harvested 48 h after transfection as previously described. Cells were washed twice with ice-cold PBS and harvested by centrifugation for 20 min at 200 g and 4 °C. Pellets were re-suspended in RIPA lysis buffer (150 mM NaCl, 50 mM Tris/HCl pH 8, 1% NP-40, 0.5% sodium deoxycholate, 0.1% SDS) supplemented with complete protease inhibitor cocktail (Roche). Lysates were incubated for 10 min on ice and subsequently centrifuged for 10 min at 13.000 g and 4 °C. 5x SDS-PAGE protein loading buffer was added to the supernatant. The samples were then incubated at 95 °C for 5 minutes. Tris-Glycine SDS-PAGE and wet immunoblotting were performed for subsequent analysis. Antibodies used were anti-mouse FLAG (1:2000, Sigma, F3165), anti-goat Actin HRP (1:5000, Jackson, 805-035-180) and anti-mouse HRP (1:5000, abcam, ab97040).

## GTEX analysis of ADAR1 and ADAR2 targets

To explore the effect of mismatches in specific locations within the dsRNA structure on ADAR1-mediated editing levels, we retrieved a list of Alu elements from the human genome using the Repeatmasker[71] track of annotation downloaded from the UCSC genome browser[72]. We then filtered for pairs of Alu elements that were (1) located in UTRs, (2) oppositely oriented, (3) within 2000 nt from each other, and (4) lacked any other Alu elements in the window. The dsRNA structures of the Alu pairs were predicted using the FOLD[73] program from the RNAStructure package[73] with default parameters. The bpRNA tool was used to assign structural elements such as loop, bulge and stems to each nucleotide in the structure[74]. The editing level for each adenosine in those structures was calculated using pooled reads from the GTEX database[75] (9125 samples in total). Only adenosines exhibiting editing levels above 3% were considered for further analysis. We then considered a window of 80 nt centered around each editing site, and annotated each position within this window—for each editing site—as being either closed (stem structure) or open (loop or bulge). For each position within this window, we then divided all editing sites into ones that harbor an open structure at that position, versus a closed one, and calculated the difference in mean editing levels. Statistical significance was determined using a t-test.

For ADAR2 analysis on the basis of the GTEX dataset, we began with 17 editing sites identified as ADAR2 targets in mice[28]. We were able to identify to map and retrieve editing levels on the basis of[4] for 10 of the 17 targets. Genomic sequences of different lengths (500, 1000, 1500, 2000, and 3000 bp) from both sides of the editing sites were extracted, and the dsRNA structures were predicted using the FOLD program from the RNAStructure package[73] with default parameters. Structures that included the highest number of additional editing sites previously described in these genomic locations[4] with the lowest free energy were chosen for further analysis. As a minimal support for the relevance of the structure, we required that at least two editing sites be found across the predicted structure, retaining 8 of the 10 targets. In our analyses, we considered only the most highly edited site in each structure, and filtered out one case where this site was at the end of a stem/loop structure (and where the +26 position was a position predicted to base pair with a position *upstream* of the site). The bpRNA tool[74] was used to assign structural elements such as loop, bulge and stems to each nucleotide in the structure.

## Target RNA editing by recruiting exogenous ADAR2 using plasmid-born arRNAs

**Plasmid construction.** Gene fragments containing arRNAs and KpnI sites were ordered from Twist Bioscience. All sequences were KpnI-digested and cloned into the digested EPB104 backbone (Addgene plasmid # 68369) with transcription of arRNA driven by a U6 promoter. The list of TWIST gene fragments is described in Supplementary Data 1. Additionally, the pDECKO-mCherry plasmids expressing 'Positive ctrl', 'Empty Vector' and 'Mismatch 35' arRNAs were retrieved from[49].

**Transient transfections.** $5 \times 10^5$ cells were plated on a 6-well plate so that cells reached 70–90% confluency at the time of transfection. 24 hours after cell seeding, 1ug of ADAR1- or ADAR2-expressing pcDNA3.1(-) plasmid, 0.1 ug of pEGFP-N1 plasmid (for assessment of transfection efficiency), and 3ug of the corresponding arRNA-expressing plasmid were transfected according to Lipofectamine® 2000 DNA Transfection Reagent Protocol. 24 hours later, the medium was changed, and 12 hours later, cells were harvested.

**RNA processing and editing quantification.** RNA isolation, DNase digestion, and reverse transcription were performed using NucleoZOL (Macherey-Nagel), Amplification Grade DNase I (Thermo Fisher Scientific), and MultiScribe Reverse Transcriptase cDNA synthesis kit

(Thermo Fisher Scientific), respectively. The subsequent PCR with KAPA HiFi HotStart ReadyMix (Roche) was performed using transcript-specific primers (Supplementary Data 1). Finally, A-to-I editing within the target mRNA was determined via Sanger sequencing (Supplementary Data 1) and the quantitative analysis using the EditR tool[76] and MultiEditR[77].

**GAPDH amplicon library preparation and analysis of sequencing data.** Editing elicited by GAPDH-targeting arRNAs was quantified using Amplicon Illumina Sequencing. Total RNA was poly-A selected using oligo dT-beads (Dynabeads mRNA DIRECT Kit life tech), and DNase-treated (Thermo Fisher Scientific). The target UTR editing region was reverse transcribed, PCR amplified (primers in Supplementary Data 1), and sequenced on the Illumina Novaseq platform. Data was analyzed by a custom R script. Reads containing wrong starting and ending sequences, and GAPDH-unaligned reads were filtered out. The editing percentage was quantified as $(G/(A + G)) *100$ at the target adenosine position.

## Reporting summary

Further information on research design is available in the Nature Portfolio Reporting Summary linked to this article.

## Data availability

The NGS data generated in this study have been deposited in the NCBI BioProject database under accession code ID PRJNA943413. The raw data regarding expression of B2 and mNG constructs in WT HEK293T cells[49] used in this study are available in the Gene Expression Omnibus database under accession code GSE155490. For the analysis of ADAR1 and ADAR2 endogenous targets, data were retrieved from the GTEX database[75].

## Code availability

The R script for analyzing NGS sequencing data from the B2 and mNG oligo libraries is accessible as Supplementary Data 2.

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

## Acknowledgements
On behalf of S.S., this project has been funded by the Israel Science Foundation (ISF) under grant 913/21-[Schwartz]. Additionally, E.Y.L. received support from the Foundation Fighting Blindness and the ISF.

## Author contributions
S.S., M.Z., and E.Y.L. conceived the study. M.Z. and M.W. performed the experiments with valuable assistance from A.U. and R.N. Data analysis was carried out by M.Z., S.S., Z.R., and E.Y.L., with support from A.U. The manuscript was collaboratively written by S.S., M.Z., and M.W., with significant contributions from A.U., R.N., S.B.A., and E.Y.L.

## Competing interests
The authors declare no competing interests.
