## [Peer Review File · Nature Communications]

Dissecting the basis for differential substrate specificity of ADAR1 and ADAR2.REVIEWER COMMENTS

Reviewer #1 (Remarks to the Author):

In the submitted manuscript, the authors reported their study on the substrate selectivity of ADAR1 and ADAR2, which deaminate double-stranded RNAs to diversify the transcriptome and modulate the immunogenicity of self and foreign transcripts. There is substantial interest in site-directed RNA editing today, so the study would be of interest to the readership of Nature Communications. However, there are some open questions that the authors should address.

Major comments:

- 1) Fig 1 - Expression levels seem to make a difference (endogenous vs overexpressed ADAR1). Hence, are the higher level of edits by ADAR2 simply due to higher ADAR2 expression? Also, different factors can affect the expression level. For example, have the authors compared biological replicates, where there is more likely to be variability in transfection efficiency than technical replicates? How about different promoters of variable strengths to drive ADAR1/2 expression?
- 2) Although the authors are saying that editing is induced 26nt upstream of a structural disruption, there seems to be some range in Fig 2A. The heatmap also shows that certain adenosines are more prone to editing, given that there are columns of increased redness within the two dotted diagonal lines.
- 3) Fig 3 - Can the authors try 3 RBDs in ADAR2? Does the offset increase? Here, it is odd to be testing ADAR homologs with only 1 or 2 RBDs when the same paragraph hypothesized that "the addition of a third RBD might give rise to an increased increment of -35."
- 4) All experiments are done in HEK293T. Are the offsets a peculiarity in HEK293T? More broadly, how generalizable are the results (-35 and -26nt for ADAR1 and ADAR2 respectively)? Is there any evidence from endogenous transcripts? There must be many imperfect dsRNAs with bulges and mismatches.
- 5) The authors created a section on the use of ADAR2 in targeted editing. However, reliance on endogenous ADAR2 for therapeutic applications is unlikely because ADAR2 is lowly expressed in my cell types/ tissues. Additionally, the authors use exogenously expressed ADAR2, but there is endogenous ADAR1 in almost all biological contexts - why bother with this ADAR2? Moreover, the authors wrote: "In some clinical contexts, it could potentially be beneficial to induce editing only in cells expressing one of the two ADAR enzymes." Like?
- 6) Fig 4E-F: The claim of reducing off-target editing due to the authors' finding is unconvincing. This is basically the same strategy published by others where a 'G' is placed opposite the cis off-target adenosine.
- 7) Fig 5A - 'G' is not enriched downstream of editing site for ADAR1/ADAR2. This goes against what is

known about the human ADARs. Does this suggest that synthetic data are not representative of the endogenous situation?

8) Fig 5B: No A-U in the data?

9) "Introduction of A-A or A-G mismatches opposite of the edited site both substantially decreased editing at the targeted position and gave rise to increased editing at a -26 bp offset" - As mentioned, papers have been published using a 'G' opposite a cis off-target site. In those published data, was there an extra off-target induced 26bp away? (Or 35bp if ADAR1.)

10) Fig 5C-F: ADAR1 or ADAR2?

11) Fig 5E-F: When the G-A/G-G/3nt mismatch is created, wouldn't it trigger off-target editing 26nt/35nt away?

Minor comments:

12) Fig 1A vs 1C: The authors should be consistent in their nomenclature. Top arm = upper arm, and bottom arm = lower arm?

13) Were the tested ADAR homologs from *Suricata suricatta* and *Octopus vulgaris* ADAR1 or ADAR2?

14) "...we designed ADAR-recruiting RNAs (arRNA) to elicit editing on four distinct endogenous targets ..." Should be five, not four.

15) Fig 4D - Mismatch-35 can also increase ADAR2-editing. Does this mean it can be challenging to selectively use one ADAR but not the other?

16) "...when the cytidine opposite to 'G' at position -1 is mismatched with a guanosine, and even more so with an adenosine ..." This phrase sounds odd. Why is cytidine mismatched with a guanosine? Presumably the authors mean "... when the nucleotide opposite 'G' at position -1 is a guanosine, and even more so an adenosine ..."?

17) Fig 5D - suggest including a schematic to illustrate.

Reviewer #2 (Remarks to the Author):

Zambrano-Mila et al., used synthetic sequences to systematically identify the preferential secondary structure for ADAR2 editing. In addition, authors demonstrated that using this new discovered secondary structure allows improved design of ADAR2-recruiting therapeutics, with increased on-target

editing efficiency. Overall, it is an interesting study with most conclusions supported by experimental data. Below are specific comments for the authors to address:

1. In Fig 2, the authors showed that ADAR2-mediated editing is induced at a constant interval of 26 bp upstream from structural disruptions. I believed that the edited adenosine is in canonical Watson-crick base-pair in this context. What would be the editing efficiency if the adenosine is in A-G, A-C or A-A base-pair? Whether the base-pair nucleotide opposite to adenosine affects the editing preference of this interval of 26 bp?
2. In Fig 3C, the mutant ADAR2-RBDs_ADAR1-deaminase and ADAR1-RBDs_ADAR2-deaminase showed only about 5% differential editing level at the preferential downstream position, which is very low. Could be this low editing due to no binding of mutant fusion ADAR proteins to RNA substrates? If this is the case, it is not convincing that the authors could use the mutant to show that RBD is the determinant for the offset position. In addition, is it possible that deaminase is the determinant for the offset distance since both ADAR2 RBD1 deaminase and ADAR2 RBD2 deaminase showed a peak at 26nt upstream as well?
3. In Fig 4, authors demonstrated that engineered ADAR2-recruiting RNA had least off-target effect by showing a lower editing at 27bp upstream of a 3-bp mismatch. However, did author examine presence of editing along the target RNA sequence further up/downstream or outside of the complementary region? Would the engineered ADAR2-recruiting RNA induce any editing along or outside the complementary region?
4. Fig 5D&E showed that G-A mismatch at -1 position gave better editing efficiency for GA sequence. Did authors test arRNA with combination of -1 G-A mismatch and 3bp mismatch at +26bp and whether this combination induces synergistic effect on this editing?
5. ADAR2 seems to be more well-tolerated with the G at -1 position as shown in Fig S6. Is this difference due to intrinsic property of different ADAR proteins or due to higher ADAR2 protein expression level?
6. Authors showed that ADAR1 and ADAR2 have each preferential offset distance from structure disruption to editing sites. How does this relate to the known endogenous ADAR1 or ADAR2 specific substrate editing?

Reviewer #3 (Remarks to the Author):

In this manuscript, Zambrano-Mila et al. systematically designed and constructed a pool of sequence variants, based on previous study from the same lab (Mol Cell 2021, PMID: 33905683, which deciphering editing specificity of ADAR1), to probe the targeting specificity of ADAR2, a homolog of ADAR1. Through comparison of ADAR1 and ADAR2, this study revealed that the two enzymes both induce A-to-I editing in a symmetric and orientation-specific way, while differ in the size of the offset from the structural disruption. ADAR2 gives rise to editing at a fixed offset of 26nt while ADAR1 induces editing in a 35nt offset as previously reported. Different offsets make possible selective editing by either ADAR1 or ADAR2. However, a mild increase in the ADAR2-mediated editing at position -26 and the coupled downstream off-target effect calls for a deeper understanding of the context-specificity of ADARs and

rationally designed structural disruptions within arRNAs. Notably, the study put forward an “offset-enhanced” model, which further deciphered the editing specificity of ADAR2, thus opening up more possibilities for improved ADAR2-recruiting therapeutic strategies.

However, a major concern is that the observed differences in the editing selectivity between ADAR1 and ADAR2 are solely based on ADARs ectopic overexpression in HEK293T cells. Thus, it still lacks direct evidence of endogenous ADARs in different type of cells or in vivo context. Collectively, the idea is cutting-edge and useful, while its advance in ADAR-targeting specificity and ADAR-recruiting therapeutics need to be further confirmed with additional cell lines and/or in vivo context.

Major concerns:

1. In Figure 1G, the boxplots suggest that number of editing events per molecule in ADAR2-OE cells is higher than those in ADAR1-OE cells, which leads to the conclusion that ADAR2 OE yields higher levels of editing compared with ADAR1, in accordance with previous studies. Did the authors calculate the average editing ratio of each editing sites per molecule? The editing level should be evaluated by the combination of the total editing events within each molecule, and the mean editing ratio of each editing event. In Figure 2B, the increase in editing levels at position -26 following ADAR2 OE is lower than the increase at position -35 following ADAR1 OE. Hence, it remains elusive why ADAR2 tends to edit more frequently in a single substrate while less efficiently in a single site with a constant offset upstream from structural disruptions. Thus, the expression level of different ectopic ADARs as well as subcellular localizations should be carefully examined in these panels to exclude the effect of unbalanced artificial ADARs overexpression.

2. In Figure 3, the authors applied several experiments, including diverse mutants, domain-swaps, and ADAR evolutionary homologs, and revealed that differences in the size of the offset between ADAR1 and ADAR2 are independent of the number of RBDs but appear to be an inherent property which can be encoded within a single RBD. Did the authors evaluate the offset preference when randomly arrange and combine ADAR1_RBDs and ADAR2_RBDs in the same enzyme? In addition, it remains unsolved whether ADAR1 and ADAR2 still retain their substrate specificities if overexpressing these two enzymes in ADAR1-KO HEK293T cells. Will ADAR1 and ADAR2 function independently, or will they induce editing in an antagonistic way due to, for example, steric hindrance?

3. In Figure 3, RBD swapping assays indicated editing is induced at ADAR-dependent fixed intervals upstream from structural disruptions. The authors concluded that RBDs primarily determine editing offsets, although the size of offsets might not scale linearly with the number of RBDs. However, in Figure 3B and 3C, it is noteworthy that the Δ editing level of ‘ADAR1-RBDs_ADAR2-deaminase’ and ‘ADAR2-RBDs_ADAR1-deaminase’ did not fully recapitulate the editing patterns observed for ADAR1 and ADAR2. The variation suggested that besides RNA recognition and binding, RBDs also play an essential role in editing efficiency, presumably by functioning synergistically with deaminase domains. Additional structural predication or analysis may be required to explain how RBDs from the ADARs holoenzyme are involved in dictating the editing capability.

4. In Figure 4, the authors chose four distinct endogenous targets harboring distinct consensus motifs to explore whether arRNAs, designed on the basis of the -26nt rule of ADAR2, could improve the efficiency of ADAR2-mediated targeted editing. However, it is not clear why the authors select these targets for examination. Any explanation? Of note, editing within ORF or UTR may derive from different modes of ADAR-mediated editing, referring to as site-specific editing and promiscuous editing, respectively. Is the -26nt rule of ADAR2-mediated editing functional equally in both modes? The authors may like to discuss this point a little bit.

We would like to begin by thanking all three reviewers for their deep engagement with our manuscript, their recognition of the importance of our findings, and the thoughtful comments that they raised. Below we list our responses (original text in *italic*, followed by our responses in plain text).

Response to Reviewer #1:

Reviewer #1: *In the submitted manuscript, the authors reported their study on the substrate selectivity of ADAR1 and ADAR2, which deaminate double-stranded RNAs to diversify the transcriptome and modulate the immunogenicity of self and foreign transcripts. There is substantial interest in site-directed RNA editing today, so the study would be of interest to the readership of Nature Communications. However, there are some open questions that the authors should address.*

Major comments:

1) *Fig 1 - Expression levels seem to make a difference (endogenous vs overexpressed ADAR1). Hence, are the higher level of edits by ADAR2 simply due to higher ADAR2 expression? Also, different factors can affect the expression level. For example, have the authors compared biological replicates, where there is more likely to be variability in transfection efficiency than technical replicates? How about different promoters of variable strengths to drive ADAR1/2 expression?*

The reviewer raises an important point, concerning the *absolute* difference in editing levels following introduction of ADAR1 vs ADAR2 into cells. We would like to emphasize that such differences - while mentioned in passing - are not at the focus of our manuscript, among others because the system that we employ (relying on overexpression of the two enzymes within cells) is relatively poorly suited for direct monitoring of such enzyme-substrate dependencies. An *in vitro* system, where activity of identical amounts of proteins can be compared, without complications due to differences in RNA levels, protein levels, localization and post-translational modifications, would be far better suited. Instead, the focus of this manuscript is on *relative differences* in editing levels. As we demonstrate in Fig. 1, as well as in the figure below, the relative editing levels achieved by endogenous vs overexpressed ADAR1 are very similar, whereas the relative editing levels achieved following ADAR1 overexpression are substantially different from the ADAR2 counterparts. These differences reflect differences in the relative substrate selectivity of these two enzymes, which this manuscript sets out to dissect, and revealing (among others) the differences in offsets with respect to structural disruptions.

Nonetheless, to more comprehensively address this comment we have meanwhile conducted a Western blot, revealing that in our system ADAR2 protein is expressed at higher levels than ADAR1, and thus suggesting that the differences in absolute levels might be a consequence of the higher expression levels. We have updated the manuscript with these results. Given that we use the same plasmid backbone for these analyses, these results might hint that ADAR2 protein is more stable than ADAR1.

Western blot of RIPA cell lysates from ADAR KO cells overexpressing FLAG-tagged ADAR1 and ADAR2, respectively. Lysates were harvested 48h after transfection with Lipofectamine 2000. Antibodies used were anti-mouse FLAG (1:2000, Sigma, F3165), anti-goat Aktin HRP (1:5000, Jackson, 805-035-180) and anti-mouse HRP (1:5000, abcam, ab97040).

To address the question regarding biological replicates: over the course of our studies we have obtained independent measurements of editing levels across our constructs across 3 *biological* replicates following ADAR2 overexpression, and across 2 biological replicates following ADAR1 overexpression. For each of these biological replicates we obtained two technical replicates (yielding 6 and 4 measurements following ADAR1 and ADAR2 overexpression, respectively). The increased editing levels following ADAR2 overexpression, in comparison to ADAR1, was consistently observed (and is consistent with the above Western blots), suggesting that this is unlikely to be driven by random fluctuations in transfection efficiencies. This figure furthermore highlights the clear difference in relative editing levels between ADAR1 and ADAR2.

Pairwise correlation coefficient matrix across biological and technical replicates, on the basis of editing levels measured in B2 perfect double-stranded constructs. The measurements were obtained from cells overexpressing either ADAR1 or ADAR2. Each cell in the heatmap represents the pairwise Spearman correlation coefficient between a pair of replicates. The color-gradient in the heatmap reflects the magnitude of the correlation coefficients.

2) *Although the authors are saying that editing is induced 26nt upstream of a structural disruption, there seems to be some range in Fig 2A.*

Yes, the editing increase driven by structural disruptions spans an interval. Across position 23-31, the median delta editing values were positive, and these values peaked at position +26. We now clarify this in the manuscript.

The heatmap also shows that certain adenosines are more prone to editing, given that there are columns of increased redness within the two dotted diagonal lines.

We completely agree that the extent of induction is not only a consequence of distance, but also of additional factors. Indeed, motivated among others by these observations, in the final set of analyses (**Fig. 5**) we set out to dissect how differences in sequence impact susceptibility to undergo editing.

3) *Fig 3 - Can the authors try 3 RBDs in ADAR2? Does the offset increase? Here, it is odd to be testing ADAR homologs with only 1 or 2 RBDs when the same paragraph hypothesized that "the addition of a third RBD might give rise to an increased increment of -35."*

In **Fig. 3C (right panel)** we perform precisely this analysis, wherein we introduce all three ADAR1 RBDs into ADAR2. We find that by doing so, the 26nt offset of ADAR2 increases to the ADAR1-typical-offset of 35. We agree that the text, as stated, had been confusing, and now reword it to "Under such a scenario, one or two RBDs might invariably give rise to an offset of -26, but never to an offset of -35, for which a third RBD would be required". We hope that this phrasing now makes it clear that the goal of these experiments was to test whether having 3 RBDs was strictly required for achieving a 35 nt offset, which is ruled out by these experiments.

4) *All experiments are done in HEK293T. Are the offsets a peculiarity in HEK293T? More broadly, how generalizable are the results (-35 and -26nt for ADAR1 and ADAR2 respectively)? Is there any evidence from endogenous transcripts? There must be many imperfect dsRNAs with bulges and mismatches.*

The reviewer raises excellent points. First, the offset-dependent editing of ADAR1 was addressed by us in the past across a panel of cell lines by Uzonyi, 2021. In all of them, endogenously expressed ADAR1 exhibited an editing induction at an offset of 35 bp, suggesting that these results are a general property of the enzyme. Such experiments could readily be conducted for ADAR1, given that ADAR2 is very lowly expressed in many cell lines and hence its activity is negligible, but is more challenging to perform for ADAR2, as it can only be monitored in cells in which all ADAR1 isoforms are knocked out.

Motivated by this comment, we have now conducted two lines of analyses aiming to find evidence for offset-mediated editing at endogenous targets in human tissues. In a first line of analysis, we first assembled a database of all endogenously-edited sites within transposable elements whose secondary structure we could readily resolve. Of note, assembling such a dataset is far from trivial: Although the overwhelming majority of edited sites occur within double stranded structures formed by two transposable elements of opposing orientations, in the majority of cases it is difficult to predict the precise secondary structure of the target site, as doing so requires knowledge about the identity of the interacting pair of transposable elements. In most cases multiple transposable elements within diverse orientations are within relatively close proximity to each other, and hence one cannot make assumptions about which pair structurally interacts and drives editing (indeed, in many cases editing might be a

consequence of heterogeneous interactions between multiple transposable elements). To overcome this challenge, we limited our analysis to a subset of 624 edited transposable elements stringently selected based on having only a single transposable element of opposite orientation in close proximity. We then used FOLD (Reuter and Mathews 2010) to predict the secondary structures of these elements and retrieved editing levels for each adenosine forming part of the structures. Editing levels were calculated based on surveying the entirety of the GTEX dataset (spanning 9125 samples from 47 tissues, collected from 548 donors). We considered adenosine site as edited by ADAR if editing levels were above 3%. On the basis of this collection, we next annotated each of the 40nt flanking each editing site as being either closed (if it resided in a stem) or open (if it resided in a loop or in a bulge). Finally, we generated an editing metaplot, wherein we calculated the extent to which structuredness within a window of 80-nt centered around an editing site impacted editing levels. Remarkably, this analysis revealed two regions in which an open structure led to increased editing: A major window 35 nt downstream of the edited site and a more minor window 30 nt upstream of the edited site. This analysis is thus fully consistent with our analyses on ADAR1, and demonstrates that the -35/30 nt rule is not only discernible in synthetic constructs in cell lines, but also apparent in an unbiased analysis in tissue-derived RNA. We furthermore found these results to be reproducible also when confining them to specific tissues, again pointing at their generality.

Difference in editing levels (Y axis) between sets of structured sequences in the indicated positions with respect to the edited site (X axis) are predicted to be open versus double-stranded. Statistical significance is color-coded as indicated. Positions -30 and +35 are highlighted in gray.

In the above analysis, we did not observe evidence for offset mediated editing also at position -26, which is consistent with the fact that most targets within transposable elements are mediated by ADAR1, rather than by ADAR2. Drastically fewer editing sites dependent exclusively on ADAR2 were characterized, and thus it was not possible to conduct the above analysis for ADAR2-associated editing sites. To assess whether we could find evidence for offset mediated editing of ADAR2, we considered a set of 10 sequences which harbor highly curated well-established targets of ADAR2 (Chalk et al. 2019). Among those sequences prediction of secondary structure could be established with high confidence for 7 sequences. An examination of ADAR2 mediated editing sites within these targets revealed that in 5 of the 7 most structures (in FLNA, GRM4, FLNB, and NOVA1), the highly edited target harbored a mismatch 26 nt away. Moreover, in the cases of GRIA4, a mismatch was present at position +27, whereas in SON it was present at position +25. While this evidence is more anecdotal in nature, it again points at the generality of our results and their relevance to editing at endogenous targets.

We have updated the manuscript with these two analyses as Figures Figure S7A-S7B.

5) *The authors created a section on the use of ADAR2 in targeted editing. However, reliance on endogenous ADAR2 for therapeutic applications is unlikely because ADAR2 is lowly expressed in my cell types/ tissues. Additionally, the authors use exogenously expressed ADAR2, but there is endogenous ADAR1 in almost all biological contexts - why bother with this ADAR2? Moreover, the authors wrote: "In some clinical contexts, it could potentially be beneficial to induce editing only in cells expressing one of the two ADAR enzymes." Like?*

We are grateful for the opportunity to better explain ourselves. Indeed, ADAR1 is typically broadly expressed, whereas ADAR2 is more cell type specific, and primarily expressed in

brain tissues. As such, ADAR2-based targeted editing offers an approach for achieving selectivity towards neural tissues. Such selectivity can either be beneficial in the context of genetic diseases manifesting in brain disorders, or potentially also in brain-derived tumors where ADAR2 expression can serve as a mechanism for achieving selectivity in targeting. We have now updated the manuscript to briefly touch on this point.

6) Fig 4E-F: The claim of reducing off-target editing due to the authors' finding is unconvincing. This is basically the same strategy published by others where a 'G' is placed opposite the cis off-target adenosine.

We agree that the reduction in off-target deamination is likely caused by the structural mismatch introduced in Fig. 4E, consistent with previous studies. The dimension that our study adds is that a rationally engineered arRNA - designed to increase editing at a fixed offset - can simultaneously also serve to decrease off-target editing. Our study illustrates how introduction of such a mismatch specifically at position 26 leads to the dual advantages of (1) increasing on-target editing ~26 nt away, and (2) reducing off-target editing at position -26.

7) Fig 5A - 'G' is not enriched downstream of editing site for ADAR1/ADAR2. This goes against what is known about the human ADARs. Does this suggest that synthetic data are not representative of the endogenous situation?

Our data suggests that the synthetic system that we interrogate captures a substantial extent of the signal that is present at endogenous sequences, and yet that the system - and in particular the experimental design - has its limitations in particular when interrogating aspects pertaining to sequence (rather than structure). The strongest sequence motif associated with editing is a depletion of G from the position upstream of the edited site. In a detailed analysis by the Bass lab (Eggington, Greene, and Bass 2011), it was found that G was the least preferred nucleotide at this position, followed by C, followed by A and U. These results are all reproduced in **Fig. 5A** (left panel), suggesting that our system is informative and captures biologically relevant attributes of A to I editing. Yet, as noted by the reviewer, different studies have consistently pointed at an enrichment of guanosines downstream of the edited site, which is not recapitulated here. This likely reflects the limitations of our design. The synthetic library that we designed was designed primarily to perturb structure, not sequence. While it systematically perturbs structure, allowing to evaluate the relevance of every base pair across the sequence, it does not comprehensively sample the sequence space. Based on this reviewer's comment, we now closely examined AG dinucleotides in both B2 and mNG. Across both constructs there were 29 monitored AG dinucleotides. We further noted, however, that in

11 of these 29 cases (38%) the AG dinucleotide happened to be preceded by a G, forming a GAG trinucleotide. While a G downstream of the edited site is beneficial for editing, an upstream G is prohibitive. As a consequence, the overall signal - depicted in the right panel of Fig. 5A - averages over both types of sequences, leading to the relatively low levels depicted in this figure. Thus, the relatively low - and imperfectly balanced - sequence space that was sampled using our approach therefore likely underlies our inability to observe an enrichment in G downstream of the edited site. We now better refer to the limitations of our study in exploring sequence in the Discussion section.

8) *Fig 5B: No A-U in the data?*

Fig 5B intends to explain how introducing mismatches affects the editing levels in the opposite A and the neighboring As. The A-U data cannot be shown since the Delta editing metric calculates the difference of editing levels between each mismatch-containing construct and perfect double-stranded construct. A-U data would describe a horizontal line on y (delta editing) =0. We now better refer to Delta editing metrics in the corresponding Figure legend to clarify this point.

9) *"Introduction of A-A or A-G mismatches opposite of the edited site both substantially decreased editing at the targeted position and gave rise to increased editing at a -26 bp offset" - As mentioned, papers have been published using a 'G' opposite a cis off-target site. In those published data, was there an extra off-target induced 26bp away? (Or 35bp if ADAR1.)*

We sought to investigate this, on the basis of the published literature, but unfortunately were unable to get our hands on relevant data. A key paper, reporting the use of 'G' opposite of cis off-target sites is (Qu et al. 2019). However, in these experiments conducted in this paper (e.g. on the KRAS gene), an A:G mismatch was introduced at all adenosines, hence precluding the ability to test whether a mismatch was induced at a 26 nt offset. A second relevant paper is one by Yi and colleagues, relying on the introduction of circular ADAR-recruiting RNAs (Yi et al. 2022). In this paper the authors had also relied on A:G mismatches to reduce off targets in the TP53 gene. Unfortunately, the raw data was not available for download, precluding an analysis of editing patterns at 26 nt offset.

10) *Fig 5C-F: ADAR1 or ADAR2?*

Fig 5C-F refers to ADAR2. The "ADAR2" term was added to the legend in each subfigure.

11) Fig 5E-F: When the G-A/G-G/3nt mismatch is created, wouldn't it trigger off-target editing 26nt/35nt away?

Excellent point. We have now reanalyzed the Sanger sequencing traces, and indeed note an off-target editing site 25 nt upstream of the single-nucleotide mismatch site and 26 bp from the 3 bp mismatch. The editing efficiency at this site is higher in all 3 samples harboring structural mismatches (G-G, G-A mismatch, or 3 nt mismatches), than in the perfect dsRNA guide, lacking such mismatches. We have added this analysis as Supplementary Figure S6C. This analysis demonstrates that the rules, inferred in this study, are not only of utility for predicting on-target effects, but also of off-target ones.

Minor comments:

12) Fig 1A vs 1C: The authors should be consistent in their nomenclature. Top arm = upper arm, and bottom arm = lower arm?

We now consistently use “upper” and “lower” arm throughout the manuscript.

13) Were the tested ADAR homologs from *Suricata suricatta* and *Octopus vulgaris* ADAR1 or ADAR2?

The sequences are annotated as being ‘ADAR’ homologs, without an indication of whether they cluster with ADAR1 or ADAR2. To examine this, we now assembled a set of ADAR homologs from across selected vertebrates and invertebrates. Sequences of the deaminase domain were retrieved using UNIPROT, aligned using CLUSTALW, and neighbor joining was used for reconstructing phylogenetic trees, with *tadA* from *E. coli* being used to root the tree.

This analysis revealed that the Suricata and Octopus proteins cluster within the 'ADAR1' part of the tree. We have updated the manuscript accordingly.

Phylogenetic tree, based on deaminase domain harboring sequences retrieved from UNIPROT and subsequently aligned using CLUSTALW. Neighbor-joining was used for reconstructing phylogenetic trees. tadA from E. coli was used as an outgroup to define the tree topology.

14) "...we designed ADAR-recruiting RNAs (arRNA) to elicit editing on four distinct endogenous targets ..." Should be five, not four.

We designed five arRNAs to elicit editing on five different target sites which are located on four different transcripts. We have now clarified this in the manuscript.

15) Fig 4D - Mismatch-35 can also increase ADAR2-editing. Does this mean it can be challenging to selectively use one ADAR but not the other?

Yes, a subtle (in this case non-statistically significant) increase is observed in cells expressing ADAR2. Yet, the effect was substantially more pronounced (and significant) in ADAR1-expressing cells. This suggests that offsets can be engineered to preferentially lead to editing by a specific deaminase, but we agree that in some cases this may prove to be challenging.

16) "...when the cytidine opposite to 'G' at position -1 is mismatched with a guanosine, and even more so with an adenosine ..." This phrase sounds odd. Why is cytidine mismatched with a guanosine? Presumably the authors mean " ... when the nucleotide opposite 'G' at position -1 is a guanosine, and even more so an adenosine ..."?

Indeed, the original phrasing of this sentence was confusing. We have now rephrased this statement as follows: "This analysis revealed that editing at adenosines in a 'GA' context (underlined A is edited) tends to be substantially higher when the nucleotide opposite of the 'G' at position -1 is a guanosine, and even more so an adenosine".

17) Fig 5D - suggest including a schematic to illustrate.

We have now updated this figure with a schematic.

Response to Reviewer #2:

Reviewer #2: *Zambrano-Mila et al., used synthetic sequences to systematically identify the preferential secondary structure for ADAR2 editing. In addition, authors demonstrated that using this new discovered secondary structure allows improved design of ADAR2-recruiting therapeutics, with increased on-target editing efficiency. Overall, it is an interesting study with most conclusions supported by experimental data. Below are specific comments for the authors to address:*

1. In Fig 2, the authors showed that ADAR2-mediated editing is induced at a constant interval of 26 bp upstream from structural disruptions. I believed that the edited adenosine is in canonical Watson-crick base-pair in this context. What would be the editing efficiency if the adenosine is in A-G, A-C or A-A base-pair? Whether the base-pair nucleotide opposite to adenosine affects the editing preference of this interval of 26 bp?

This is an interesting question, which our current design is unable to address. In our design we systematically mutated the structures (first four figures) and the sequence opposite of the edited site (as shown in Fig. 5) but we did not design sequences interrogating the combination of the two. Given that editing efficiencies are likely a product of both sequence and structure, simultaneously perturbing both will allow us to understand to what extent these two dimensions act in an additive or potentially also in a synergistic manner. This marks an interesting area for future investigation, which we have now added to the **Discussion**.

2. In Fig 3C, the mutant ADAR2-RBDs_ADAR1-deaminase and ADAR1-RBDs_ADAR2-deaminase showed only about 5% differential editing level at the preferential downstream position, which is very low. Could be this low editing due to no binding of mutant fusion ADAR proteins to RNA substrates? If this is the case, it is not convincing that the authors could use the mutant to show that RBD is the determinant for the offset position.

The reviewer is correct in pointing out that the two ADAR mutants with the RBD swaps have considerably lower levels of activity than their WT counterparts, leading to a lower dynamic range of editing levels and hence also to a decreased dynamic range of the delta-editing events that are quantified in Fig. 3C. Nonetheless, we believe the results - in particular for ADAR1_RBDs_ADAR2_deaminase are compelling, displaying a single, clear prominent peak at position -35 (Fig. 3C, right panel). It is critical to emphasize here that our findings pertaining to structural disruptions (1) do not rely on quantifying absolute editing levels, and (2) do not rely on comparison of signal between two different constructs (e.g. WT and mutant). Instead, our findings exclusively rely on quantification of *relative* signal, and this signal is always

internally normalized with respect to structurally unperturbed sequences. Therefore, our analyses should be robust to absolute differences in editing levels between different ADAR mutants.

We should furthermore highlight (as detailed below) that we had removed a large number of additional ADAR mutants that we had generated, because they failed to give rise to deamination activity. The set of mutants that we do display reflects a carefully selected subset where the levels of editing that were achieved were sufficiently robust to be informative.

In addition, is it possible that deaminase is the determinant for the offset distance since both ADAR2 RBD1 deaminase and ADAR2 RBD2 deaminase showed a peak at 26nt upstream as well?

We believe that our data is more consistent with a model wherein these offsets are accounted for by the RBDs, yet we cannot fully rule out a model wherein the deaminase domain also contributes to this offset. The fact that the RBD swapping experiments results in a swapping of the offsets suggests that the offsets are determined by the RBDs. Yet, a model that would, in principle, be consistent with our measurements is that there is a 'default' offset of 26 nt that is encoded directly by the deaminase domain, and that one (or more) of the ADAR1 RBDs contribute towards increasing this offset to 35 nt. We have now sought to test this hypothesis, by introducing constructs encoding only the ADAR1 and ADAR2 deaminase domains (without any of the RBDs), to assess whether a 26 nt offset was still observed. Unfortunately, neither of these two constructs displayed editing activity, preventing us from getting an answer. In general, perturbing ADAR enzymes while retaining function proved - as a rule - to be challenging. In addition to the set of constructs that we presented in the manuscript, we generated 13 additional ADAR variants, designed to test hypotheses regarding the length of the linkers potentially impacting offset size, and regarding the relative role of each of the 3 ADAR1 RBDs and each of the two ADAR2 RBDs in determining offset size, which failed to give rise to deamination activity (data not shown). Thus, the possibility raised by this reviewer remains a viable one, and we now discuss it in the Discussion section.

3. In Fig 4, authors demonstrated that engineered ADAR2-recruiting RNA had least off-target effect by showing a lower editing at 27bp upstream of a 3-bp mismatch. However, did author examine presence of editing along the target RNA sequence further up/downstream or outside of the complementary region? Would the engineered ADAR2-recruiting RNA induce any editing along or outside the complementary region?

The readout for the editing levels was high-throughput sequencing, for which we had amplified an amplicon surrounding the target site, and subjected it to paired end sequencing. Our sequencing results (which were limited in read length) allowed us to assess editing levels across a total of 14 adenosines spanning positions -75 to +37, with respect to the editing site, but excluding a window from -45 to -6, which was not covered by either of the two reads we had sequenced. Thus, the interrogated region spans a considerable fraction of the complementary region, but does not interrogate at all a region beyond it. We do not expect that editing would be induced beyond the complementary region, yet agree that off-target editing both within the same transcript in regions beyond the complementarity sequence ('off-targets in cis') and potentially also in other transcripts ('off-targets in trans') would be important to rule out for any clinical product. We consider our results primarily proof-of-concept results, demonstrating the potential to mitigate off-target results via improved understanding of the rules conferring specificity to the two ADAR enzymes.

4. Fig 5D&E showed that G-A mismatch at -1 position gave better editing efficiency for GA sequence. Did authors test arRNA with combination of -1 G-A mismatch and 3bp mismatch at +26bp and whether this combination induces synergistic effect on this editing?

This is an interesting suggestion, addressing the question of combinatorics, in this case between a mismatch at position -1 and one at position +26, both of which we independently find to contribute to editing. We are currently working towards establishing an approach that will allow us to simultaneously monitor the editing inducing ability of thousands of arRNAs in parallel, via which we hope to be able to systematically explore the interplay between sequence, structural mismatches at a distance, and mismatches at the edited site. However, given the challenges this imposes, we will not be able to resolve this question in the context of the current manuscript, which focuses on establishing that offset-mediated editing occurs, and in establishing proof of principle experiments demonstrating that it can be exploited for clinical targeting.

5. ADAR2 seems to be more well-tolerated with the G at -1 position as shown in Fig S6. Is this difference due to intrinsic property of different ADAR proteins or due to higher ADAR2 protein expression level?

We believe that a definitive answer to this question will require in-vitro experimentation, allowing comparison of activity between purified components. We believe that when expressed within cells, there are too many regulatory levels at the transfection, transcriptional, translational and post-translational levels that can confound this kind of analysis. We have

meanwhile confirmed that, as anticipated by this reviewer, ADAR2 is expressed at higher levels in our system, which could underlie the difference in absolute signal. Nonetheless, given that the colored-gradients represent the internally normalized delta editing levels and not absolute editing levels, this suggests to us that these differences reflect differences in the intrinsic properties of the two ADAR proteins. However, comparisons among ADARs should be taken with caution because of the limited A numbers.

Western blot of RIPA cell lysates from ADAR KO cells overexpressing FLAG-tagged ADAR1 and ADAR2, respectively. Lysates were harvested 48h after transfection with Lipofectamine 2000. Antibodies used were anti-mouse FLAG (1:2000, Sigma, F3165), anti-goat Aktin HRP (1:5000, Jackson, 805-035-180) and anti-mouse HRP (1:5000, abcam, ab97040).

6. Authors showed that ADAR1 and ADAR2 have each preferential offset distance from structure disruption to editing sites. How does this relate to the known endogenous ADAR1 or ADAR2 specific substrate editing?

Motivated by this comment, we have now conducted two lines of analyses aiming to find evidence for offset-mediated editing at endogenous targets in human tissues. In a first line of analysis, we first assembled a database of all endogenously edited sites within transposable elements whose secondary structure we could readily resolve. Of note, assembling such a dataset is far from trivial: Although the overwhelming majority of edited sites occur within double-stranded structures formed by two transposable elements of opposing orientations, in the majority of cases it is difficult to predict the precise secondary structure of the target site, as doing so requires knowledge about the identity of the interacting pair of transposable elements. In most cases, multiple transposable elements within diverse orientations are within relatively close proximity to each other, and hence one cannot make assumptions about which pair structurally interacts and drives editing (indeed, in many cases editing might be a consequence of heterogeneous interactions between multiple transposable elements). To overcome this challenge, we limited our analysis to a subset of 624 edited transposable elements stringently selected based on having only a single transposable element of opposite orientation in close proximity. We then used FOLD (Reuter and Mathews 2010) to predict the secondary structures of these elements and retrieved editing levels for each adenosine forming part of the structures. Editing levels were calculated based on surveying the entirety of the GTEX dataset (spanning 9125 samples from 47 tissues, collected from 548 donors).

We considered adenosine site as edited by ADAR if editing levels were above 3%. On the basis of this collection, we next annotated each of the 40nt flanking each editing site as being either closed (if it resided in a stem) or open (if it resided in a loop or in a bulge). Finally, we generated an editing metaplot, wherein we calculated the extent to which structuredness within a window of 80-nt centered around an editing site impacted editing levels. Remarkably, this analysis revealed two regions in which an open structure led to increased editing: A major window 35 nt downstream of the edited site and a more minor window 30 nt upstream of the edited site. This analysis is thus fully consistent with our analyses on ADAR1, and demonstrates that the -35/30 nt rule is not only discernible in synthetic constructs in cell lines, but also apparent in an unbiased analysis in tissue-derived RNA. We furthermore found these results to be reproducible also when confining them to specific tissues (not shown), again pointing at their generality.

Difference in editing levels (Y axis) between sets of structured sequences in the indicated positions with respect to the edited site (X axis) are predicted to be open versus double-stranded. Statistical significance is color-coded as indicated. Positions -30 and +35 are highlighted in gray.

In the above analysis we did not observe evidence for offset mediated editing also at position -26, which is consistent with the fact that most targets within transposable elements are mediated by ADAR1, rather than by ADAR2. Drastically fewer editing sites dependent exclusively on ADAR2 were characterized, and thus it was not possible to conduct the above analysis for ADAR2-associated editing sites. To assess whether we could find evidence for offset mediated editing of ADAR2, we considered a set of 10 sequences which harbor highly curated well-established targets of ADAR2(Chalk et al. 2019) . Among those sequences

prediction of secondary structure could be established with high confidence for 7 sequences. An examination of ADAR2-mediated editing sites within these targets revealed that in 5 of the 7 most structures (in FLNA, GRM4, FLNB, and NOVA1), the highly edited target harbored a mismatch 26 nt away. Moreover, in the cases of GRIA4, a mismatch was present at position +27, whereas in SON it was present at position +25. While this evidence is more anecdotal in nature, it again points at the generality of our results and their relevance to editing at endogenous targets.

We have updated the manuscript with these two analyses as Figures Figure S7A-S7B.

Response to Reviewer #3:

Reviewer #3: *In this manuscript, Zambrano-Mila et al. systematically designed and constructed a pool of sequence variants, based on previous study from the same lab (Mol Cell 2021, PMID: 33905683, which deciphering editing specificity of ADAR1), to probe the targeting specificity of ADAR2, a homolog of ADAR1. Through comparison of ADAR1 and ADAR2, this study revealed that the two enzymes both induce A-to-I editing in a symmetric and orientation-specific way, while differ in the size of the offset from the structural disruption. ADAR2 gives rise to editing at a fixed offset of 26nt while ADAR1 induces editing in a 35nt offset as previously reported. Different offsets make possible selective editing by either ADAR1 or ADAR2. However, a mild increase in the ADAR2-mediated editing at position -26 and the coupled downstream off-target effect calls for a deeper understanding of the context-specificity of ADARs and rationally designed structural disruptions within arRNAs. Notably, the study put forward an “offset-enhanced” model, which further deciphered the editing specificity of ADAR2, thus opening up more possibilities for improved ADAR2-recruiting therapeutic strategies.*

However, a major concern is that the observed differences in the editing selectivity between ADAR1 and ADAR2 are solely based on ADARs ectopic overexpression in HEK293T cells. Thus, it still lacks direct evidence of endogenous ADARs in different type of cells or in vivo context.

In point 2, below, we now present two analyses, confirming the relevance of the rules inferred here to editing at endogenous sites within human tissues. These analyses confirm the general relevance of our findings, and - to a considerable level - address this important concern.

Collectively, the idea is cutting-edge and useful, while its advance in ADAR-targeting specificity and ADAR-recruiting therapeutics need to be further confirmed with additional cell lines and/or in vivo context.

Major concerns:

1. In Figure 1G, the boxplots suggest that number of editing events per molecule in ADAR2-OE cells is higher than those in ADAR1-OE cells, which leads to the conclusion that ADAR2 OE yields higher levels of editing compared with ADAR1, in accordance with previous studies. Did the authors calculate the average editing ratio of each editing sites per molecule? The editing level should be evaluated by the combination of the total editing events within each molecule, and the mean editing ratio of each editing event.

We are not entirely sure we understood this comment. Figure 1G portrays the number of A-G editing events across single molecules, all derived from a single DNA species (either the mNG WT sequence on the left, or B2 on the right). At the single molecule level, each editing event is binary, i.e. an adenosine is either edited into a G, or unedited. What we are now presenting, below - and which we hope addresses the comment raised by the reviewer - is a distribution of the mean editing levels (i.e. averaged across all molecules) of all adenosines across the B2 and mNG genes, following overexpression of either ADAR1 or ADAR2. Consistent with the results displayed based on the single-molecule analysis presented in the manuscript, the graph below confirms generally higher levels of editing following ADAR2 overexpression in comparison to ADAR1.

In Figure 2B, the increase in editing levels at position -26 following ADAR2 OE is lower than the increase at position -35 following ADAR1 OE. Hence, it remains elusive why ADAR2 tends to edit more frequently in a single substrate while less efficiently in a single site with a constant offset upstream from structural disruptions. Thus, the expression level of different ectopic ADARs as well as subcellular localizations should be carefully examined in these panels to exclude the effect of unbalanced artificial ADARs overexpression.

The reviewer raises an important point, concerning the *absolute* difference in editing levels following introduction of ADAR1 vs ADAR2 into cells. We would like to emphasize that such differences - while mentioned in passing - are not at the focus of our manuscript, among others because the system that we employ relying on overexpression of the two enzymes within cells is relatively poorly suited for direct monitoring of protein activity. An *in vitro* system, where activity of identical amounts of proteins can be compared, without complications due to localization and post-translational modifications, would be far better suited. Instead, the focus

of this manuscript is on *relative differences* in editing levels. As we demonstrate in Fig. 1, the relative editing levels achieved by endogenous vs overexpressed ADAR1 are nearly identical, whereas the relative editing levels achieved following ADAR1 overexpression are substantially different from the ADAR2 counterparts. These differences reflect differences in the substrate selectivity of these two enzymes, which this manuscript sets out to dissect, and revealing (among others) the differences in offsets with respect to structural disruptions.

Nonetheless, to more comprehensively address this comment we have meanwhile quantified ADAR1 and ADAR2 protein levels, revealing that in our system ADAR2 protein is expressed at higher levels than ADAR1, and thus suggesting that the differences in absolute levels might be a consequence of the higher expression levels. We have updated the manuscript with these results. Given that we use the same plasmid backbone for these analyses, these results might hint that ADAR2 protein is more stable than ADAR1.

Western blot of RIPA cell lysates from ADAR KO cells overexpressing FLAG-tagged ADAR1 and ADAR2, respectively. Lysates were harvested 48h after transfection with Lipofectamine 2000. Antibodies used were anti-mouse FLAG (1:2000, Sigma, F3165), anti-goat Aktin HRP (1:5000, Jackson, 805-035-180) and anti-mouse HRP (1:5000, abcam, ab97040).

The reviewer further notes, astutely, that while in our hands ADAR2 gives rise to higher *absolute* editing levels, the *relative* effect of the offset is substantially more pronounced for ADAR1 than for ADAR2. We fully agree with this observation, and can think of different explanations that might rationalize it. One explanation might be that the mechanism achieving selectivity in targeting is completely decoupled from the mechanism underlying the deamination activity. Under such a model, each of the two can be tuned, independently, to different levels. A second possibility might be that the higher deamination activity of ADAR2 leads to (partial) saturation of its targets, and hence limits their ability to become additionally induced via structural offsets. We tend to favor the former model, among others based on the consideration that some of the ADAR2 variants explored in this manuscript have low levels of activity, yet still retain relatively low levels of offset-mediated editing suggesting that the system has not reached saturation.

2. In Figure 3, the authors applied several experiments, including diverse mutants, domain-swaps, and ADAR evolutionary homologs, and revealed that differences in the size of the offset between ADAR1 and ADAR2 are independent of the number of RBDs but appear to be an inherent property which can be encoded within a single RBD. Did the authors evaluate the offset preference when randomly arrange and combine ADAR1_RBDs and ADAR2_RBDs in the same enzyme?

This experiment is, conceptually, a very interesting one, but based on our experience might be challenging to implement. In addition to the four mutants of ADAR1 and ADAR2 that we published in the paper, we have attempted to generate 13 additional variants of the enzymes, exploring additional hypothesis pertaining to their offset mediated activity. In these constructs we sought to explore the role of each of the individual RBDs of ADAR1 and ADAR2 in giving rise to offset mediated editing, as well as to explore the role played by the size of the linker between the RBDs and the deaminase domain. These variants - despite our best attempts to carefully design them, while taking into account available structural information - all failed to give rise to detectable levels of editing (and hence are not shown in the manuscript). On the basis of this relatively low success rate, an experiment involving random rearrangements and combinations might prove to be informative, but only if it can be conducted and interrogated at a sufficient scale that can overcome a high failure rate.

In addition, it remains unsolved whether ADAR1 and ADAR2 still retain their substrate specificities if overexpressing these two enzymes in ADAR1-KO HEK293T cells. Will ADAR1 and ADAR2 function independently, or will they induce editing in an antagonistic way due to, for example, steric hindrance?

In this manuscript we have opted for a reductionist approach, in which the activity of each of these two enzymes is monitored in the context where the other is not expressed. The tacit assumption is that these enzymes operate independently, and are not mutually dependent on each other. The following considerations are supportive of this assumption: (1) In many cells and tissue types, ADAR2 is expressed at negligible levels, and hence all editing activity is likely entirely mediated by ADAR1, (2) the literature on these two enzymes consistently points at some substrates being exclusively ADAR1 substrates, while others are exclusively ADAR2 substrates (while others, yet, are shared) pointing at the separability between the activity of the two enzymes (Yang et al. 1997; Burns et al. 1997; Melcher et al. 1996; Riedmann et al. 2008). (3) the relative editing levels achieved following overexpression of ADAR1 correlate well with editing levels in cells expressing ADAR1 endogenously. These considerations all

point at the relevance of our findings, and help to rule out the possibility that our reductionist system may not be representative of the more complex settings within cells.

Nonetheless, the extent to which lessons learned from synthetic constructs in perturbed cell lines do leave open the question as to the extent to which these are relevant for editing at endogenous targets in tissues. To address this broader question, we have now conducted two lines of analyses aiming to find evidence for offset-mediated editing at endogenous targets in human tissues. In a first line of analysis, we first assembled a database of all endogenously-edited sites within transposable elements whose secondary structure we could readily resolve. Of note, assembling such a dataset is far from trivial: Although the overwhelming majority of edited sites occur within double stranded structures formed by two transposable elements of opposing orientations, in the majority of cases it is difficult to predict the precise secondary structure of the target site, as doing so requires knowledge about the identity of the interacting pair of transposable elements. In most cases multiple transposable elements within diverse orientations are within relatively close proximity to each other, and hence one cannot make assumptions about which pair structurally interacts and drives editing (indeed, in many cases editing might be a consequence of heterogeneous interactions between multiple transposable elements). To overcome this challenge, we limited our analysis to a subset of 624 edited transposable elements stringently selected based on having only a single transposable element of opposite orientation in close proximity. We then used FOLD (Reuter and Mathews 2010) to predict the secondary structures of these elements and retrieved editing levels for each adenosine forming part of the structures. Editing levels were calculated based on surveying the entirety of the GTEX dataset (spanning 9125 samples from 47 tissues, collected from 548 donors). We considered adenosine site as edited by ADAR if editing levels were above 3%. On the basis of this collection, we next annotated each of the 40nt flanking each editing site as being either closed (if it resided in a stem) or open (if it resided in a loop or in a bulge). Finally, we generated an editing metaplot, wherein we calculated the extent to which structuredness within a window of 80-nt centered around an editing site impacted editing levels. Remarkably, this analysis revealed two regions in which an open structure led to increased editing: A major window 35 nt downstream of the edited site and a more minor window 30 nt upstream of the edited site. This analysis is thus fully consistent with our analyses on ADAR1, and demonstrates that the -35/30 nt rule is not only discernible in synthetic constructs in cell lines, but also apparent in an unbiased analysis in tissue-derived RNA. We furthermore found these results to be reproducible also when confining them to specific tissues (not shown), again pointing at their generality.

Difference in editing levels (Y axis) between sets of structured sequences in the indicated positions with respect to the edited site (X axis) are predicted to be open versus double-stranded. Statistical significance is color-coded as indicated. Positions -30 and +35 are highlighted in gray.

In the above analysis we did not observe evidence for offset mediated editing also at position -26, which is consistent with the fact that most targets within transposable elements are mediated by ADAR1, rather than by ADAR2. Drastically fewer editing sites dependent exclusively on ADAR2 were characterized, and thus it was not possible to conduct the above analysis for ADAR2- associated editing sites. To assess whether we could find evidence for offset mediated editing of ADAR2, we considered a set of 10 sequences which harbor highly curated well-established targets of ADAR2 (Chalk et al. 2019). Among those sequences prediction of secondary structure could be established with high confidence for 7 sequences. An examination of ADAR2 mediated editing sites within these targets revealed that in 5 of the 7 most structures (in FLNA, GRM4, FLNB, and NOVA1), the highly edited target harbored a mismatch 26 nt away. Moreover, in the cases of GRIA4, a mismatch was present at position +27, whereas in SON it was present at position +25. While this evidence is more anecdotal in nature, it again points at the generality of our results and their relevance to editing at endogenous targets.

We have updated the manuscript with these two analyses as Figures Figure S7A-S7B.

3. In Figure 3, RBD swapping assays indicated editing is induced at ADAR-dependent fixed intervals upstream from structural disruptions. The authors concluded that RBDs primarily determine editing offsets, although the size of offsets might not scale linearly with the number of RBDs. However, in Figure 3B and 3C, it is noteworthy that the Δ editing level of 'ADAR1-RBDs_ADAR2-deaminase' and 'ADAR2-RBDs_ADAR1-deaminase' did not fully recapitulate the editing patterns observed for ADAR1 and ADAR2. The variation suggested that besides RNA recognition and binding, RBDs also play an essential role in editing efficiency, presumably by functioning synergistically with deaminase domains. Additional structural predication or analysis may be required to explain how RBDs from the ADARs holoenzyme are involved in dictating the editing capability.

We completely agree with this comment and the need for structures of the full ADAR enzymes in complex with RNA in order to understand the mechanism underlying offset mediated editing. We do not see this point as being in conflict with our conclusions. Our conclusions are focused on understanding the basis for offset-mediated editing, whereas the reviewer points out that the RBDs might also play a more general role in mediating editing. In this context we are in agreement with the reviewer that the genetic approach that we adopted, while powerful, can at times - in particular when associated with globally reduced activity - also be difficult to interpret. Decreased editing levels upon perturbations of RBDs might indeed point at a direct role of RBDs in modulating deamination levels, as suggested by the reviewer. Of relevance to this point is previous literature, revealing a role for the ADAR dsRBDs in bringing the deaminase domain in close proximity to the editing site (Stefl et al. 2010), in assisting in base flipping by increasing the conformational flexibility of nucleotides in the duplex adjacent to its binding site (Yi-Brunozzi, Stephens, and Beal 2001), and in helping stabilize dimerization of ADAR2 catalytic domains (Thuy-Boun et al. 2020). Conversely, in our case loss of editing

activity might also be indirect, as a result of inadvertently compromising the tertiary structure of the protein.

4. In Figure 4, the authors chose four distinct endogenous targets harboring distinct consensus motifs to explore whether arRNAs, designed on the basis of the -26nt rule of ADAR2, could improve the efficiency of ADAR2-mediated targeted editing. However, it is not clear why the authors select these targets for examination. Any explanation? Of note, editing within ORF or UTR may derive from different modes of ADAR-mediated editing, referring to as site-specific editing and promiscuous editing, respectively. Is the -26nt rule of ADAR2-mediated editing functional equally in both modes? The authors may like to discuss this point a little bit.

We selected sites based on two loose considerations: (1) we selected targets within genes at which editing had previously been induced (this was mainly applied as a de-risking strategy, as we were concerned that some genes may potentially not be amenable to targeting, e.g. due to their intracellular localization), and (2) we prioritized targets in the UTR region since it has been shown that editing yields are higher in the 3'-UTR compared to ORF and 5'-UTR (Paul Vogel et al., 2018 & Tobias Merkle et al 2019).

It is also interesting to speculate as to whether and how offset-mediated editing may relate to biases in the regions that are edited (ORF vs UTR) and to 'site-specific' versus 'promiscuous' editing. The latter question relates to the question of whether, from a mechanistic perspective, 'site-specific' is distinct from 'promiscuous' editing, or whether the two are different manifestations of an identical mechanism. To address these questions, we believe it will be critical to structurally resolve ADAR1 and ADAR2 proteins in a complex with diverse RNA substrates.

References

- Burns, C. M., H. Chu, S. M. Rueter, L. K. Hutchinson, H. Canton, E. Sanders-Bush, and R. B. Emeson. 1997. "Regulation of Serotonin-2C Receptor G-Protein Coupling by RNA Editing." *Nature* 387 (6630): 303–8.
- Chalk, Alistair M., Scott Taylor, Jacki E. Heraud-Farlow, and Carl R. Walkley. 2019. "The Majority of A-to-I RNA Editing Is Not Required for Mammalian Homeostasis." *Genome Biology* 20 (1): 1–14.
- Eggington, Julie M., Tom Greene, and Brenda L. Bass. 2011. "Predicting Sites of ADAR Editing in Double-Stranded RNA." *Nature Communications* 2: 319.
- Melcher, T., S. Maas, A. Herb, R. Sprengel, P. H. Seeburg, and M. Higuchi. 1996. "A Mammalian RNA Editing Enzyme." *Nature* 379 (6564): 460–64.
- Merkle, T., Merz, S., Reautschnig, P. et al. "Precise RNA editing by recruiting endogenous ADARs with antisense oligonucleotides". *Nature Biotechnology* 37, 133–138 (2019). <https://doi.org/10.1038/s41587-019-0013-6>
- Qu, Liang, Zongyi Yi, Shiyong Zhu, Chunhui Wang, Zhongzheng Cao, Zhuo Zhou, Pengfei Yuan, et al. 2019. "Programmable RNA Editing by Recruiting Endogenous ADAR Using Engineered RNAs." *Nature Biotechnology* 37 (9): 1059–69.
- Reuter, Jessica S., and David H. Mathews. 2010. "RNAstructure: Software for RNA Secondary Structure Prediction and Analysis." *BMC Bioinformatics* 11 (March): 129.
- Riedmann, Eva M., Sandy Schopoff, Jochen C. Hartner, and Michael F. Jantsch. 2008. "Specificity of ADAR-Mediated RNA Editing in Newly Identified Targets." *RNA* 14 (6): 1110–18.
- Stefl, Richard et al. 2010. "The Solution Structure of the ADAR2 dsRBM-RNA Complex Reveals a Sequence-Specific Readout of the Minor Groove". *Cell*, Volume 143, Issue 2, 225 - 237
- Thuy-Boun, Alexander S., Justin M. Thomas, Herra L. Grajo, Cody M. Palumbo, Sehee Park, Luan T. Nguyen, Andrew J. Fisher, and Peter A. Beal. 2020. "Asymmetric Dimerization of Adenosine Deaminase Acting on RNA Facilitates Substrate Recognition." *Nucleic Acids Research* 48 (14): 7958–72.
- Uzonyi, A. Nir, R., Shliefer, O., Stern-Ginossar, N., et al. 2021. "Deciphering the principles of the RNA editing code via large-scale systematic probing". *Molecular Cell* 81: 2374–2387.e3.
- Vogel, P., Moschref, M., Li, Q. et al. "Efficient and precise editing of endogenous transcripts with SNAP-tagged ADARs". *Nature Methods* 15, 535–538 (2018). <https://doi.org/10.1038/s41592-018-0017-z>
- Yang, J. H., P. Sklar, R. Axel, and T. Maniatis. 1997. "Purification and Characterization of a

Human RNA Adenosine Deaminase for Glutamate Receptor B Pre-mRNA Editing.” *Proceedings of the National Academy of Sciences of the United States of America* 94 (9): 4354–59.

Yi-Brunozzi, H. Y., O. M. Stephens, and P. A. Beal. 2001. “Conformational Changes That Occur during an RNA-Editing Adenosine Deamination Reaction.” *The Journal of Biological Chemistry* 276 (41): 37827–33.

Yi, Zongyi, Liang Qu, Huixian Tang, Zhiheng Liu, Ying Liu, Feng Tian, Chunhui Wang, et al. 2022. “Engineered Circular ADAR-Recruiting RNAs Increase the Efficiency and Fidelity of RNA Editing in Vitro and in Vivo.” *Nature Biotechnology* 40 (6): 946–55.

REVIEWER COMMENTS

Reviewer #1 (Remarks to the Author):

The authors have added some nice analyses and explanations to address my previous comments. I now recommend publication in the journal.

Reviewer #2 (Remarks to the Author):

The authors have well addressed all my major concerns. I have no further comments.

Reviewer #3 (Remarks to the Author):

This revised manuscript has addressed some of this reviewer's concerns, including screening ADAR2-mediated editing across systematically-designed sequence variants based on previous study, and dissecting the basis for substrate specificity between ADAR1 and ADAR2. Overall, the manuscript is well written and has provided some lines of evidence draw the conclusion.

One open question has not been directly addressed in this revised manuscript. More specifically, expression of ADAR1 largely differs from that of ADAR2 across different cell lines and tissues, as ADAR1 is ubiquitously expressed while ADAR2 is most highly expressed in brain tissues. This manuscript has applied HEK29T cells for the study, while ADAR2 is barely expressed in HEK29T. The authors have specified that editing patterns in synthetic constructs following ADAR1 overexpression were correlated well with those in WT HEK293T cells with endogenously expressed ADAR1. This conclusion is reasonable. However, it is questionable whether overexpressing ADAR2 in HEK293T could fully recapitulate the editing patterns of ADAR2, as ADAR2 is barely expressed in HEK293T. One clear limitation of this study is to solely using ADAR ectopic overexpression to address the editing selectivity differences between ADAR1 and ADAR2 in HEK293T cells, which however have different endogenous ADAR1 and ADAR2 expression background. The authors may like to test their finding/conclusion in other cell lines that have higher endogenous ADAR2 expression.

Response to reviewers:

We are also grateful to the reviewers for their continuous engagement with our manuscript. Below we address the single remaining comment raised by reviewer #3 (original comment in *black italics*, our response in blue).

This revised manuscript has addressed some of this reviewer's concerns, including screening ADAR2-mediated editing across systematically-designed sequence variants based on previous study, and dissecting the basis for substrate specificity between ADAR1 and ADAR2. Overall, the manuscript is well written and has provided some lines of evidence draw the conclusion.

We are delighted that the reviewer is overall appreciative of the extensive revision that we had conducted.

One open question has not been directly addressed in this revised manuscript. More specifically, expression of ADAR1 largely differs from that of ADAR2 across different cell lines and tissues, as ADAR1 is ubiquitously expressed while ADAR2 is most highly expressed in brain tissues. This manuscript has applied HEK29T cells for the study, while ADAR2 is barely expressed in HEK29T. The authors have specified that editing patterns in synthetic constructs following ADAR1 overexpression were correlated well with those in WT HEK293T cells with endogenously expressed ADAR1. This conclusion is reasonable. However, it is questionable whether overexpressing ADAR2 in HEK293T could fully recapitulate the editing patterns of ADAR2, as ADAR2 is barely expressed in HEK293T. One clear limitation of this study is to solely using ADAR ectopic overexpression to address the editing selectivity differences between ADAR1 and ADAR2 in HEK293T cells, which however have different endogenous ADAR1 and ADAR2 expression background. The authors may like to test their finding/conclusion in other cell lines that have higher endogenous ADAR2 expression.

The reviewer is raising the concern here that ADAR2 levels achieved in our system - following ADAR2 overexpression - may potentially not reflect physiological ADAR2 levels within tissues/cells. In our view, such a concern would have been a very valid one, had our goal been to use such an overexpression system in order to characterize the editing landscapes transcriptome wide. In this case, it is well possible that overexpression of ADAR2 would lead to editing activity at sites that are not naturally edited. However, this is not the goal of the manuscript. Instead, the goal was to compare the substrate specificity of ADAR1 to its ADAR2 counterpart: to understand how and why these two enzymes have different targets. For this endeavor, the reductionist approach that we employ is perfectly suited, allowing side-by-side comparison of activity of the two enzymes, on an identical set of substrates, and under an identical experimental regime.

Moreover, we are concerned that conducting the experiment, as proposed by the reviewer, would not be informative. Cells naturally expressing ADAR2 invariably also express ADAR1, and hence any readout within such a system will be a convolution of editing by the two enzymes. Our approach was carefully designed precisely in order to overcome this shortcoming, allowing us to cleanly distinguish ADAR1 from ADAR2 activity.

We should moreover emphasize that in the context of the revision we have also been able to show the relevance of the rules inferred in this study to sites detected in human tissues, both for ADAR1 and for ADAR2. These results demonstrate that the rules inferred from our reductionist approach are at work also under the most physiological contexts imaginable (i.e. human tissues), and in this sense go above and beyond what can be learned from an additional set of experiments in a transformed cell line.

To summarize, we don't believe the experiment suggested by the reviewer is required to support the findings of the manuscript, nor that it can be performed in a 'clean' way, and we therefore believe it is unlikely that it will provide much - if any - added value to the manuscript.